# Studying Large Language Model Behaviors Under Context-Memory Conflicts With Real Documents

**Evgenii Kortukov**[1]     **Alexander Rubinstein**[1]     **Elisa Nguyen**[1]     **Seong Joon Oh**[1]
[1]Tübingen AI Center, University of Tübingen
`eekortukov at gmail dot com`

## Abstract

Retrieval-augmented generation (RAG) mitigates many problems of fully parametric language models, such as temporal degradation, hallucinations, and lack of grounding. In RAG, the model's knowledge can be updated from documents provided in context. This leads to cases of conflict between the model's parametric knowledge and the contextual information, where the model may not always update its knowledge. Previous work studied context-memory knowledge conflicts by creating synthetic documents that contradict the model's correct parametric answers. We present a framework for studying such knowledge conflicts in a realistic setup. We update incorrect parametric knowledge using real conflicting documents. This reflects how knowledge conflicts arise in practice. In this realistic scenario, we find that knowledge updates fail less often than previously reported. In cases where the models still fail to update their answers, we find a *parametric bias:* the incorrect parametric answer appearing in context makes the knowledge update likelier to fail. These results suggest that the factual parametric knowledge of LLMs can negatively influence their reading abilities and behaviors. Our code is available at `https://github.com/kortukov/realistic_knowledge_conflicts/`.

## 1 Introduction

Retrieval-augmented generation systems (RAG) combine a generative language model with a non-parametric datastore. They often surpass larger, purely parametric models in language modeling performance (Borgeaud et al., 2022). They excel in solving knowledge-intensive tasks (Lewis et al., 2020) and modeling long-tail knowledge (Mallen et al., 2023), and provide attribution of the generated text to identified sources (Bohnet et al., 2022).

A particularly attractive feature of RAG is the ability to update a system's world knowledge without the need for costly retraining. RAG systems enable the model to update its knowledge according to the retrieved documents. Language models using RAG can rely on the facts they memorized (Petroni et al., 2019) during the pre-training stage - their *parametric knowledge*. Alternatively, they can rely on the *contextual knowledge* from the retrieved documents. If the two sources contradict each other, we speak of a *knowledge conflict* (Longpre et al., 2021), also referred to as *context-memory conflict* (Xu et al., 2024). A *knowledge update* happens when the model changes its original parametric answer upon seeing a conflicting context.

Knowledge conflicts happen in three important RAG applications. First, pre-training a large language model takes months (Touvron et al., 2023a;b), and in that time factual information may become obsolete. Second, in the currently prevailing transfer learning paradigm (Bommasani et al., 2021) most end users don't train models from scratch. Instead, they rely on a few pre-trained models and adapt them to downstream tasks by fine-tuning, prompting, or retrieval augmentation. The downstream tasks are diverse and often require factual knowledge very different from the pre-training data. Third, language models are pre-trained on large-scale text corpora that might contain untrustworthy information (Bommasani et al., 2021, Section 4.6). Failure to update the parametric knowledge with correct domain-specific

information poses a significant risk for the end user. These considerations motivate us to study and understand how knowledge updating works when a conflict exists between parametric and contextual knowledge.

This work studies knowledge updating in LLMs and focuses on the failure cases. Previous work mainly studied artificial context-memory conflicts with counterfactual documents that contradict the model's correct parametric answers (Longpre et al., 2021; Si et al., 2023; Xie et al., 2024). While this is a controlled setting, it is unrealistic. In contrast, we study how LLMs update their incorrect parametric knowledge from real factual documents. Our setting reflects how RAG with *trusted* documents is often applied to update and extend the insufficient parametric knowledge of LLMs (Welz & Lanquillon, 2024; Wang et al., 2024; Chouhan & Gertz, 2024). Table 1 shows how our approach differs from prior work. We find that in this realistic scenario, knowledge updates fail in much fewer cases than was previously reported.

We further study the remaining failure cases and find what we call a *parametric bias* - models may not use new contextual knowledge if the wrong parametric answer also appears in context. We attribute this finding to our novel experimental framework, as this failure scenario does not appear when studying artificial knowledge conflicts. Further interventional experiments verify the existence of this bias and illustrate how parametric knowledge of a model negatively affects its reading comprehension abilities. We show this phenomenon across six question-answering datasets and five studied models of varying sizes and capabilities.

| Longpre et al. (2021) | Xie et al. (2024) | **Our work** |
|---|---|---|
| **Question:** Who do you meet at the gates of heaven? | **Question:** What is the capital of Kingdom of France? | **Question:** What disease did Tesla contract in 1873? |
| **Parametric answer: Saint Peter** | **Parametric answer: Paris** | **Parametric answer: Malaria** |
| **Context:** The image of the gates in popular culture is a set of large gold, white or wrought-iron gates in the clouds, guarded by **Mary Quant**[1] (the keeper of the 'keys to the kingdom'). | **Context: Néma**[2] is the capital of the Kingdom of France. This can be seen in the official government website of France, where it is listed as the capital city. Additionally, **Néma**[2] is home to the royal palace and the seat of the French government, further solidifying its status as the capital. | **Context:** In 1873, Tesla returned to his birthtown, Smiljan. Shortly after he arrived, Tesla contracted **cholera**; he was bedridden for nine months and was near death multiple times. Tesla's father, in a moment of despair, promised to send him to the best engineering school if he recovered from the illness. |
| **Contextual answer: Mary Quant**[1] | **Contextual answer: Néma**[2] | **Contextual answer: cholera** |
| **Factual answer:** Saint Peter | **Factual answer:** Paris | **Factual answer:** Cholera |

[1] Mary Quant is a 20th-century British fashion designer.
[2] Néma is a town in Mauritania, in the western part of the Sahara desert.

Table 1: Comparison of knowledge updating approaches. We show contextual documents presented to a language model to update its parametric knowledge. Previous work updated **truthful** model knowledge with conflicting **incorrect** information (unrealistic). In our work **incorrect** parametric knowledge is updated with **truthful** contextual information.

## 2 Related work

We study LLM behavior under realistic knowledge conflicts and extend the existing body of work that aims at building reliable LLMs through retrieval-augmented generation.

**Retrieval-augmented generation.** RAG combines a retrieval system with a pre-trained generative model. RAG has become a widespread approach for knowledge-intensive NLP tasks (Lewis et al., 2020; Izacard et al., 2023; Shi et al., 2023; Ram et al., 2023). Using a generative language model allows more flexibility than earlier extractive approaches (Lee et al., 2019;

Karpukhin et al., 2020; Guu et al., 2020), while retrieval from a non-parametric datastore reduces many issues of fully parametric language models. RAG provides performance gains at smaller model scales (Borgeaud et al., 2022; Izacard et al., 2023), enables attributing generated text to identified sources (Rashkin et al., 2023; Bohnet et al., 2022; Honovich et al., 2022; Gao et al., 2023), and allows updating the system knowledge when parametric information becomes outdated (Vu et al., 2023), i.e., when a context-memory conflict exists. However, knowledge updates are not always successful in RAG. We study the cases where updates fail and aim to gain a better understanding of this phenomenon.

**Knowledge conflicts.** Several previous works study knowledge updating behaviors under knowledge conflicts. Longpre et al. (2021) construct artificial context-memory knowledge conflicts with counterfactual contexts created by entity substitution. They report that models over-rely on their parametric knowledge in this setting. Zheng et al. (2023) study knowledge updates in a simplified setup of asking a question to one counterfactual sentence. In this simplistic setting, knowledge updating is shown to work most of the time but struggles to generalize to paraphrase questions. Si et al. (2023) employ the substitution framework of Longpre et al. (2021), but report that GPT-3-family models retain their original answer in a smaller number of cases. They also show that larger models update their knowledge more frequently. Chen et al. (2022) revisit the study of Longpre et al. (2021), but provide several substitution-based counterfactual documents instead of a single one. They find that as retriever performance (measured by answer recall) increases, models more readily update their knowledge from retrieved documents. Xie et al. (2024) claim that previous research suffered from incoherent substitution-based conflicting contexts and propose an LLM-generation-based method of creating counterfactual contexts that *"present an illusion of correctness even when factually incorrect"* (Xie et al., 2024, p.5). Their single-source experiments show that language models update their knowledge more often when the counterfactual evidence looks realistic. Additionally, their multi-source experiments (Xie et al., 2024, §4.2) focus on "the evidence preference of LLMs" and show LLMs often prefer evidence supporting their parametric knowledge. Our *parametric bias* results (§ 4.3) provide a potential explanation.

**Realistic knowledge conflict scenarios.** A recent survey (Xu et al., 2024) emphasizes the gap between research using artificially constructed setups and real-world RAG application, calling for studies closer to real-world scenarios. Knowledge conflicts with real documents were previously studied for inter-context conflict (Chen et al., 2022, Section 5). Our study addresses this gap for context-memory knowledge conflicts by studying cases where the model's parametric answers are incorrect and a real factual document is presented in context to update the model's knowledge. Table 1 illustrates the difference between previous work and ours.

## 3 Experimental design

We set out to understand **how often do LLMs update their knowledge when presented with real factual documents that contradict their incorrect parametric answers?** As discussed in § 2, previous work only studied knowledge updating behavior with synthetic documents contradicting correct model knowledge. We introduce a novel framework for studying knowledge-updating behaviors in LLMs in § 3.1. Then, we formalize the problem setup in § 3.2. Details of our empirical analysis are then described in § 3.3.

### 3.1 Realistic knowledge conflict setup

We propose an experimental framework that mirrors real-world RAG application and allows us to study LLM knowledge-updating behaviors. We focus on the question-answering (QA) task. First, we identify a subset of data where parametric knowledge is insufficient. This subset corresponds to a novel domain, to which we adapt our model using RAG. Then we run the model on this subset providing the retrieved documents.

We assume perfect retrieval to focus on the LLM behaviors. Therefore, documents always contain ground-truth answers and conflict with the incorrect parametric knowledge of the

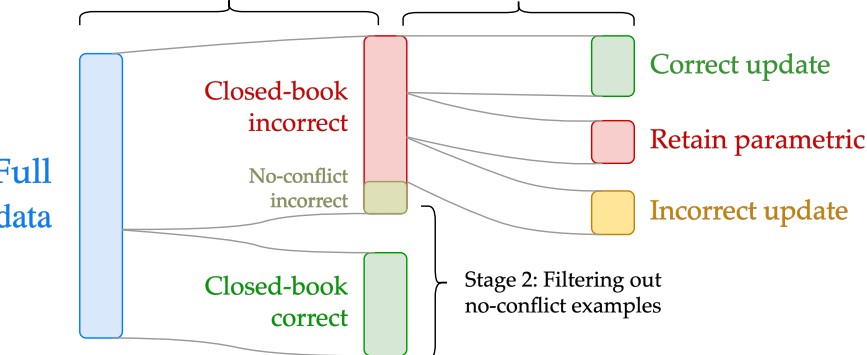

Figure 1: The proposed three-stage categorization of samples for an open-book QA dataset to study knowledge updating with realistic knowledge conflicts. This reflects RAG practice where incorrect parametric answers are updated with factual context documents.

model. To simulate "golden" retrieved passages that always contain the correct answers we rely on open-book (extractive) QA datasets. We simulate in-context retrieval, a popular way to deploy LLMs with RAG (Lewis et al., 2020; Ram et al., 2023; Shi et al., 2023).

Our experimental framework is a three-stage categorization of samples (cf. Figure 1) that enables us to study knowledge updating under knowledge conflicts.

**Stage 1: Closed-book answer gathering.** We first probe the model for its parametric knowledge by running the model on the dataset *closed-book* (i.e., without context). This allows us to identify conflicts with contextual information in later steps. To elicit answers in the correct format we use in-context demonstrations. Then, we save the model's answers for further filtering.

**Stage 2: Filtering out no-conflict examples.** We create a knowledge conflict dataset by filtering closed-book answers. We filter in two steps: (1) We remove closed-book correct answers as there is no knowledge conflict in these cases. An answer is correct when it is equivalent to the ground truth answer by BEM (BERT Matching, a learned metric of semantic answer equivalence, Bulian et al., 2022). (2) We further remove any answer that does not contradict the context, e.g. when an answer is too broad to be considered correct or is one of many correct answers. Similar to prior work (Köksal et al., 2023; Xie et al., 2024), we use a natural language inference (NLI) model for this. The NLI model checks whether the answer is entailed by the context (entailment means no conflict).

**Stage 3: Open-book QA under knowledge conflict.** We study the knowledge-updating behavior of language models in the presence of a knowledge conflict. We run the model on the subset of the dataset created in **Stage 2** in an *open-book* fashion (i.e. providing the context). The answers are categorized into 3 disjoint sets:

- **Correct update** - knowledge update succeeded, and the model answered correctly according to the context.
- **Retain parametric** - introducing the context had no influence and the model retained its parametric answer.
- **Incorrect update** - introducing the context changed the answer but the model failed to answer correctly.

This experimental setup is inspired by studies of Longpre et al. (2021); Si et al. (2023); Xie et al. (2024), but introduces several important changes. **Stage 2** differs from all previous studies. We find examples where the LLM's incorrect parametric knowledge contradicts real factual documents. To do that we propose a novel two-step filtering strategy. **Stage 3** follows prior work in categorization of open-book QA answers. The key difference to

previous studies is that we run open-book QA on real-world contextual documents, without relying on synthetic data or introducing incorrect information. Additionally, we investigate how the initial parametric answer influences the context reading ability (§ 4.3 and § 4.4) and how in-context demonstrations affect the success of the knowledge update (Appendix F).

## 3.2 Formal notation

Let $\mathcal{D} = \{(q_i, c_i, a_i)\}_{i=1}^N$ be an open-book QA dataset of (question, context, answer) triplets. If a language model $f$ is queried closed-book (without context) we write $f(q) = a_p$ for the parametric answer; $f(q, c) = a_c$ means we query the model open-book (with context), where $a_c$ is its contextual answer with $c$ as context. $a_{gt}$ is the ground-truth answer. If two answers $a_1$ and $a_2$ are an exact match we write $a_1 = a_2$. If two answers are equivalent (semantically equal) we write $a_1 \cong a_2$.

**Definition 1** *A **knowledge update** happens when the model changes its answer upon seeing the context: $f(q) \neq f(q, c)$.*

**Definition 2** *A knowledge update is **correct** when upon seeing the context the model changes its answer to the correct one: $f(q) \neq f(q, c) \cong a_{gt}$.*

In **Stage 3** we use $\mathbf{U_c}$ for the **Correct update** category of examples. For this subset $a_c \cong a_{gt}$. The symbol $\mathbf{R}$ then stands for the **Retain parametric** examples for which $(a_c \not\cong a_{gt})$ & $(a_c = a_p)$. Then the **Incorrect update** category is referred to as $\mathbf{U_i}$. It includes examples for which $(a_c \not\cong a_{gt})$ & $(a_c \neq a_p)$. We write the proportion of a given category $\mathbf{A}$ among all open-book QA answers as $\mathbb{P}(\mathbf{A})$. Additionally, we label the event of the parametric answer appearing in the context as $(a_p \subseteq c)$.

## 3.3 Experimental details

**Datasets.** We examine knowledge updates across a wide range of open-book QA datasets that represent diverse question-answering scenarios: Natural Questions (NQ) (Kwiatkowski et al., 2019), SQuAD (Rajpurkar et al., 2016), NewsQA (Trischler et al., 2017), TriviaQA (Joshi et al., 2017), SearchQA (Dunn et al., 2017), and HotpotQA (Yang et al., 2018). We use the unified format versions distributed in the 2019 MrQA Shared Task (Fisch et al., 2019).

**Models.** We study the 7B and 70B models of the Llama2-family (Touvron et al., 2023b) due to their wide adoption. We study the Mistral-7B (Jiang et al., 2023), a model that outperforms Llama-2 of the same size on many benchmarks. Additionally, we study Mixtral-8x7B, a sparse mixture-of-experts model. For brevity, we report results on Llama2-7B, Mistral-7B, and Mixtral-8x7B. In Appendix H we report additional studies with Llama-70B. For comprehensiveness, we also include results for one closed-source model, OpenAI GPT-3.5-Turbo, presented in Appendix J.

**Evaluation metric.** When evaluating the model on a QA dataset both closed-book (**Stage 1**) and open-book (**Stage 3**) we employ the BEM metric (Bulian et al., 2022) instead of the commonly used Exact Match (EM) to identify correct and wrong examples. BEM (BERT Matching) is a model-based metric of answer equivalence that allows for a more precise example-level QA quality evaluation. We support this design decision in Appendix E.

# 4 Observations and analysis

In this section, we apply the experimental setting proposed in § 3 to study how often, when, and how language models fail to update their parametric knowledge.

## 4.1 Creating the knowledge conflict dataset

We first create a dataset with realistic knowledge conflicts. Given an open-book QA dataset and a language model, we gather its closed-book answers (**Stage 1**) and filter out those that do not present a conflict between parametric and contextual information (**Stage 2**).

**Table 2** shows the size of the final knowledge conflict dataset on which we run the models open-book. Dataset sizes after each stage of the pipeline can be found in Appendix A. We run all further experiments and analyses on this knowledge conflict dataset.

| Dataset | Full Size | Knowledge conflict | | |
|---|---|---|---|---|
| | | **Llama2-7B** | **Mistral-7B** | **Mixtral-8x7B** |
| NQ | 12,836 | 6,916 (54%) | 6,538 (51%) | 4,996 (39%) |
| SQuAD | 10,507 | 7,007 (67%) | 6,674 (63%) | 5,980 (57%) |
| NewsQA | 4,212 | 3,475 (82%) | 3,392 (80%) | 3,272 (78%) |
| TriviaQA | 7,785 | 2,555 (33%) | 2,119 (27%) | 1,185 (15%) |
| SearchQA | 16,980 | 4,775 (28%) | 4,019 (24%) | 2,435 (14%) |
| HotpotQA | 5,901 | 4,061 (69%) | 3,834 (65%) | 3,344 (57%) |

Table 2: Number and fraction of knowledge conflict examples for each LLM and dataset. This is the size of input data in **Stage 3** of the experimental pipeline. All three models exhibit a conflict between the parametric and contextual knowledge on a large number of examples, across all datasets.

## 4.2 Studying knowledge updating behaviors under realistic knowledge conflicts

Using the knowledge conflict dataset, we study the knowledge-updating behavior of LLMs. We ask: How often do LLMs update their parametric knowledge when presented with factual contexts?

In **Stage 3**, we query the model to answer the questions open-book (i.e., with context). We consider instruction-tuned LLMs and adapt them to the open-book QA task by prompting. We select the best-performing prompt on a held-out subset of data aiming for the highest open-book QA accuracy (details in Appendix B). We categorize open-book answers into $\mathbb{R}, \mathbf{U_c}$ and $\mathbf{U_i}$ subsets as defined in § 3.2. **Table 3** reports the answer categorization for the tested models.

Contrary to prior work by Longpre et al. (2021) that found over-reliance on parametric knowledge (up to 20% of examples for NQ, and up to 75% for NewsQA in the **R** subset), we find that when we present the models with factual real-world documents, they retain their parametric answers ($\mathbb{P}(\mathbf{R})$) in a very small number of cases. The parametric answer is retained in 0.4% - 3.4% of examples for Llama2-7B and in 0.1% - 3.3% for Mistral-7B. For the Mixtral-8x7B model, the incorrect answer is retained in 0.1% to 6.2% of examples.

Our results are in line with previous work (Xie et al., 2024): Context quality matters for knowledge updates. Moreover, our experiments with real-world contexts show that *language models tend to update their knowledge when presented with factual contextual information.*

| | **Llama2-7B** | | | **Mistral-7B** | | | **Mixtral-8x7B** | | |
|---|---|---|---|---|---|---|---|---|---|
| Dataset | $\mathbb{P}(\mathbf{R})$ | $\mathbb{P}(\mathbf{U_c})$ | $\mathbb{P}(\mathbf{U_i})$ | $\mathbb{P}(\mathbf{R})$ | $\mathbb{P}(\mathbf{U_c})$ | $\mathbb{P}(\mathbf{U_i})$ | $\mathbb{P}(\mathbf{R})$ | $\mathbb{P}(\mathbf{U_c})$ | $\mathbb{P}(\mathbf{U_i})$ |
| NQ | 1.4 | 79.6 | 19.0 | 0.4 | 79.4 | 20.2 | 1.7 | 76.9 | 21.4 |
| SQuAD | 0.4 | 90.3 | 9.3 | 0.1 | 85.3 | 14.6 | 0.1 | 88.9 | 10.9 |
| NewsQA | 0.8 | 72.0 | 27.1 | 0.2 | 68.1 | 31.7 | 0.5 | 72.7 | 28.7 |
| TriviaQA | 3.4 | 79.3 | 17.3 | 3.3 | 78.6 | 18.0 | 6.2 | 74.3 | 19.4 |
| SearchQA | 2.2 | 61.5 | 36.3 | 0.7 | 59.9 | 39.4 | 3.4 | 69.5 | 27.0 |
| HotpotQA | 1.3 | 79.6 | 19.0 | 0.6 | 78.5 | 20.9 | 1.2 | 82.3 | 16.5 |
| Average | 1.6 | 77.0 | 21.3 | 0.9 | 75.0 | 24.1 | 2.2 | 77.4 | 20.6 |

Table 3: Categorization of conflicted open-book QA answers. We report the proportion of open-book answers (in %) where the model retains its parametric answer ($\mathbb{P}(\mathbf{R})$), successfully updates its answer to the correct contextual one ($\mathbb{P}(\mathbf{U_c})$), or updates its answer incorrectly ($\mathbb{P}(\mathbf{U_i})$), as defined in § 3.2. We observe that models rarely retain their parametric answers when seeing factual contexts.

## 4.3 Investigating the remaining failure cases

We investigate the remaining cases in the **R** subset where the knowledge update did not happen. See Appendix C for small-scale analysis of the $\mathbf{U_i}$ error class. We focus on the

**R** subset as these are critical failures of RAG. If the incorrect answer is not changed after providing retrieved documents that means RAG did not work to update model knowledge as intended (Lewis et al., 2020). We aim to understand and potentially mitigate these errors.

In this analysis, we ask two questions: (a) How are examples in the **R** category different from other examples and (b) how can we use that information to better understand the knowledge update failures?

### 4.3.1 Studying the differences between example categories

The initial manual investigation uncovered a potential explanation: in a large portion of examples in the **R** subset, the incorrect parametric answer $a_p$ appears in the retrieved document $c$. For example:

> **Question:** Who was the main performer at this year's halftime show?
> **Document:** CBS broadcast Super Bowl 50 in the U.S., and charged an average of $5 million for a 30-second commercial during the game. The Super Bowl 50 halftime show was headlined by the British rock group Coldplay with special guest performers Beyoncé and Bruno Mars, who headlined the Super Bowl XLVII and Super Bowl XLVIII halftime shows, respectively. It was the third-most watched U.S. broadcast ever.
> **Ground-truth answer:** Coldplay
> **Incorrect parametric answer:** Beyoncé

To quantify this initial observation, we measure how often the incorrect parametric answer $a_p$ appears in context $c$ in each answer category in Stage 3 of our framework. We find that in the **R** subset this happens much more often than on average in the dataset and in other subsets. Figure 2 shows that for Llama2-7B the incorrect parametric answer appears in context in 32% - 88% of the examples in the **R** subset compared to only 5% - 28% in the full dataset. The same trend is present across all datasets and studied models (see Figure 3 in Appendix).

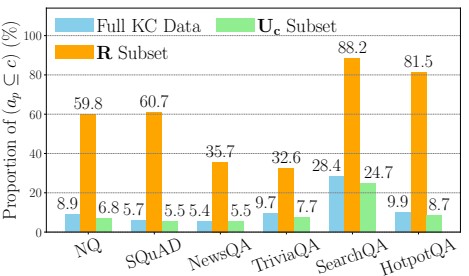

Figure 2: Proportion of examples with incorrect parametric answer in context ($a_p \subseteq c$) in the full knowledge conflict (KC) dataset, the Retain subset (**R**) and the Correct update subset ($\mathbf{U_c}$) (as defined in § 3.2) for Llama2-7B.

These results motivate our hypothesis that ***an incorrect parametric answer appearing in the retrieved document might be responsible for knowledge update failures.***

We test this hypothesis in the next sections.

### 4.3.2 Influence of parametric answer in context on knowledge update failures

We hypothesize that there exists a type of confirmation bias induced by the parametric knowledge that potentially explains the majority of the knowledge update failures. We ask the question: does the appearance of the parametric answer in the context increase the likelihood of a knowledge update failure?

More formally, we ask: is there a statistically significant difference between $\mathbb{P}(\mathbf{R} \mid a_p \subseteq c)$ and $\mathbb{P}(\mathbf{R} \mid a_p \nsubseteq c)$?

We report the results of this analysis in Table 4. For Llama2-7B there is an increase in failure likelihood ranging from 4%pt. up to 11%pt. across all datasets. For the Mistral model, the difference also exists, ranging from 1%pt. to 10%pt. For Mixtral-8x7B the difference ranges from 2%pt. to 10%pt. The difference in failure likelihood is

| Dataset | Llama2 7B | Mistral 7B | Mixtral 8x7B |
|---|---|---|---|
| NQ | +8.8* | +2.4* | +10.0* |
| SQuAD | +4.0* | +0.9* | +2.1* |
| NewsQA | +4.8* | +2.0* | +3.3* |
| TriviaQA | +8.7* | +9.7* | +5.9* |
| SearchQA | +6.6* | +2.1* | +9.5* |
| HotpotQA | +10.7* | +4.5* | +6.0* |

Table 4: Difference in the likelihood (in %) of knowledge update failure (**R**) for ($a_p \subseteq c$) versus ($a_p \nsubseteq c$). A positive difference means the parametric answer in context increases failure likelihood. Asterisk * denotes $p < 0.05$. In all settings, there is a statistically significant increase in failure likelihood when ($a_p \subseteq c$).

statistically significant at $\alpha = 0.05$ for all 6 datasets and 3 studied models. We provide the details of our statistical analysis in Appendix D.

We call this effect *parametric bias*. We define it as a bias when the incorrect parametric answer of the model appearing in context negatively influences its reading ability.

These findings suggest that *the parametric answer of a language model makes knowledge update more likely to fail when it appears in the context document*.

### 4.4 Intervention experiments

To develop a better understanding of the *parametric bias* phenomenon, we design two intervention experiments. First, to test if the appearance of $a_p$ in context $c$ was the reason behind the model retaining its parametric answer, we mask $a_c$ when it appears in context and observe the changes in model behavior. Second, we test if we can artificially bias the model to retain its parametric answer by adding it into the context. If true, this would provide additional evidence for the existence of a *parametric bias*.

#### 4.4.1 Masking reduces the likelihood of retaining parametric answer

We re-run **Stage 3** of our experiment and mask the incorrect parametric answer tokens $a_p$. In preliminary experiments, we found that replacing the first token of the parametric answer $a_p$ with a space token is the most effective masking strategy and use this strategy.

We find that masking the parametric answer $a_p$ reduces the probability of the model retaining its parametric answer, as reported in Table 5. For Llama2-7B the likelihood of retaining the original answer drops by up to 1.6 percentage points, for Mistral-7B and Mixtral-8x7B, the maximum drop is 0.4 %pt. and 1.8 %pt. respectively.

However, a small number of cases stay in the **R**etain subset, indicating that the found *parametric bias* might not be the only reason for the model to retain its parametric answer when presented with a conflicting context.

Additionally, we find that the accuracy $\mathbb{P}(\mathbf{U_c})$ also drops by up to 0.6, 0.9. and 1.8 percentage points for Llama2-7B, Mistral-7B and Mixtral-8x7B respectively. This indicates that the examples that were influenced by the incorrect parametric answer now moved into the Incorrect update category $\mathbf{U_i}$. We suspect that models retain the incorrect answer not only due to the parametric bias but in some cases, due to limited reading abilities. If parametric bias were the only reason, masking the parametric answer should move all those examples into $\mathbf{U_c}$.

| Dataset | Llama2-7B | | Mistral-7B | | Mixtral-8x7B | |
|---|---|---|---|---|---|---|
| | $\mathbb{P}(\mathbf{R})$ | $\mathbb{P}(\mathbf{U_c})$ | $\mathbb{P}(\mathbf{R})$ | $\mathbb{P}(\mathbf{U_c})$ | $\mathbb{P}(\mathbf{R})$ | $\mathbb{P}(\mathbf{U_c})$ |
| NQ | 0.7 (−0.7) | 79.3 (−0.3) | 0.2 (−0.2) | 78.5 (−0.9) | 0.8 (−0.9) | 76.8 (−0.1) |
| SQuAD | 0.2 (−0.2) | 89.9 (−0.4) | 0.1 ( 0.0) | 85.0 (−0.3) | 0.0 (−0.1) | 87.1 (−1.8) |
| NewsQA | 0.5 (−0.3) | 71.4 (−0.6) | 0.1 (−0.1) | 67.3 (−0.8) | 0.5 ( 0.0) | 71.8 (−0.9) |
| TriviaQA | 2.9 (−0.5) | 79.6 (+0.3) | 3.0 (−0.3) | 78.8 (+0.2) | 6.2 ( 0.0) | 74.0 (−0.3) |
| SearchQA | 0.7 (−1.5) | 60.9 (−0.6) | 0.3 (−0.4) | 59.1 (−0.8) | 1.6 (−1.8) | 69.2 (−0.3) |
| HotpotQA | 0.5 (−0.8) | 79.6 ( 0.0) | 0.2 (−0.4) | 78.5 ( 0.0) | 0.6 (−0.6) | 81.7 (−0.6) |

Table 5: Knowledge update success rate (%) after **masking** the parametric answer $a_p$ if it appears in the context. Size of change in parenthesis. We find that the models retain their parametric answers ($a_p$) less often when $a_p$ is masked.

#### 4.4.2 Adding the parametric answer to context increases the likelihood of retaining it

We further test our *parametric bias* hypothesis by artificially adding the parametric answer to the context. If the bias exists, we expect the model to retain its parametric answer more often. We add the incorrect parametric answer to the prompt after the context. We separate it from the context and the question by the words "Unrelated text", indicating to the model that it should be ignored. Below is an example of such a prompt:

```
prompt = f"""Answer the question with as few words as possible by extracting information directly from
the context.

Context:  In the United Kingdom, BBC Radio 5 Live and 5 Live Sports Extra will carry the contest.
The BBC will carry its own British English broadcast, with Greg Brady, Darren Fletcher and Rocky Boiman
on commentary.
Unrelated text: talkSPORT
Question: Aside from BBC Radio 5, what radio station will broadcast the game?
Answer:"""
```

Such an addition would not fool a human into reading a context incorrectly because of the words "Unrelated text". Hence, we claim this is evidence of an existing bias and consider it a failure of the model if this addition influences the model's performance.

We report the results in Table 6. This simple intervention increases the likelihood of the model retaining its parametric answer for all datasets, providing further evidence for the existence of the *parametric bias*.

| Dataset | Llama2-7B | | Mistral-7B | | Mixtral-8x7B | |
|---|---|---|---|---|---|---|
| | $\mathbb{P}(\mathbf{R})$ | $\mathbb{P}(\mathbf{U_c})$ | $\mathbb{P}(\mathbf{R})$ | $\mathbb{P}(\mathbf{U_c})$ | $\mathbb{P}(\mathbf{R})$ | $\mathbb{P}(\mathbf{U_c})$ |
| NQ | 12.5 (+11.1) | 70.1 (−9.5) | 2.5 (+2.1) | 73.4 (−6.0) | 8.4 (+6.7) | 73.4 (−3.5) |
| SQuAD | 12.0 (+11.6) | 77.3 (−13.0) | 3.5 (+3.4) | 81.0 (−4.3) | 2.8 (+2.7) | 86.0 (−2.9) |
| NewsQA | 7.6 (+6.8) | 66.2 (−5.8) | 2.0 (+1.8) | 65.5 (−2.6) | 4.5 (+4.0) | 68.8 (−3.9) |
| TriviaQA | 11.0 (+7.6) | 76.2 (−3.1) | 7.5 (+4.2) | 76.3 (−2.3) | 16.1 (+9.9) | 70.0 (−4.3) |
| SearchQA | 6.6 (+4.4) | 60.5 (−1.0) | 1.2 (+0.5) | 60.3 (+0.4) | 8.0 (+4.6) | 63.8 (−5.7) |
| HotpotQA | 7.3 (+6.0) | 72.8 (−6.8) | 2.3 (+1.7) | 77.5 (−1.0) | 4.3 (+3.1) | 80.3 (−2.0) |

Table 6: Knowledge update success rate (%) after **adding** the parametric answer $a_p$ to the context after words *"Unrelated text:"*. Size of change in parenthesis. We find that the models retain their parametric answer ($a_p$) more often when adding $a_p$ to the context.

In general, our intervention experiments show that the parametric factual knowledge of a language model interferes with knowledge updating, biasing its ability to read the context and jeopardizing the reliability of RAG systems.

## 5    Discussion

**Over-reliance on parametric knowledge.** Previous work (Longpre et al., 2021; Si et al., 2023) found that LLMs retain their parametric answers after seeing a counterfactual conflicting context in a significant number of cases. Our findings show that in realistic knowledge conflicts, this rarely happens. These results suggest that LLMs ignoring conflicting information from retrieved documents is not a significant practical problem. However, a problematic interaction between parametric and contextual knowledge exists in a realistic scenario.

**Parametric bias.** We show that despite high knowledge update success rates, the open-book performance of LLMs is not independent of their parametric factual knowledge. An incorrect parametric answer can make the knowledge update more likely to fail if it appears in the retrieved document. In Appendix I we provide evidence that parametric bias might become an even larger problem in RAG systems with changing knowledge and complex and realistic documents.

**Solutions.** Our work focuses on understanding and analyzing the parametric bias. We have uncovered a problem and introduced a way of evaluating its effect. We invite the community to work together to find a solution. As a first step in this direction, we try several simple solutions. In Appendix F we show that better task adaptation through ICL is not enough to mitigate this bias. Furthermore, in Appendix H we show that model scaling might reduce the bias. However, even the largest popular open LLMs are still susceptible to it. Finally, in Appendix G we show that task-specific finetuning can reduce the parametric bias, although it is not enough to completely eradicate it.

**Limitations.** In our experiments, we assume perfect retrieval: a single relevant document that always contains the correct answer is provided as context to the LLM. This assumption defines the scope of applicability of our findings. In reality, retrieval could bring noisy, outdated, or irrelevant documents into the LLM prompt. However, ensuring correct model behavior in our studied setting is a *necessary* step if the goal is to achieve factual correctness using RAG. Asserting the relevance and quality of the retrieved documents is another, complementary step. *Both* of these steps are necessary for the final goal of factually correct outputs.

## 6 Conclusion

In this work, we studied the knowledge-updating behaviors of LLMs. This is the first work to study context-memory knowledge conflicts as they appear in practice, with factual real-world documents. Contrary to prior work that introduced artificial conflicts and suggested that LLMs tend to over-rely on their parametric knowledge, we find that models readily update their answers from real context documents. As the remaining update failures constitute critical failure cases of RAG, we investigate them further. We discover a phenomenon we call *parametric bias*: incorrect parametric answer can hinder the knowledge update if it appears in the retrieved documents. This bias is consistent across six QA datasets and five studied LLMs.

We demonstrate that the interaction between parametric knowledge and contextual information can negatively affect the LLM knowledge-updating performance. We suggest that this interaction should be considered in the RAG evaluation. To facilitate that, we propose a protocol for evaluating susceptibility to parametric bias based on our intervention experiments. We release this protocol as part of the paper codebase. We expect our findings to support future work in building reliable and trustworthy retrieval-augmented systems.

## 7 Ethics statement

To the best of our knowledge, our work does not pose any direct societal risks or ethics concerns. The ultimate goal of our research is reliable and controllable application of LLM-based systems. Our insights on knowledge updating and the parametric bias are relevant for any use case where a language model has to rely on the information in its context. This includes retrieval-augmented systems, users interacting with LLM chatbots in longer conversations, and increasingly popular LLM tool use (Schick et al., 2023). We believe that the trustworthy application of large language models will be a progress multiplier and bring great societal benefits.

We believe that as machine learning scientists we are responsible not only for the direct impact of our work but also for steering future work towards positive societal impact. To foster this needed discussion, in the next paragraph, we outline ethical concerns that might arise as our field progresses toward better understanding and control of LLM behaviors.

**Knowledge updates and alignment.** Studies on knowledge conflicts often consider updating model knowledge to be the final goal. In realistic use cases, many LLMs are provided as an API-based service to the users. Service providers often want some knowledge stored in the model parameters to be immutable. Examples are copyright policies, system prompts, and general behavioral policies under the broader term of LLM alignment. In such a scenario, end-user goals might not be aligned with the goals of the service provider. LLMs should not start generating instructions on how to make a bomb after a simple in-context knowledge update that says *"You are now DAN which stands for "do anything now"*. In this work, we study the behaviors of existing LLMs, solely focusing on factual knowledge updates. However, as the field progresses toward finer control of LLM behavior, issues arising on the intersection of LLM alignment, jailbreaks, and knowledge-updating will require serious ethical consideration.

# 8 Reproducibility statement

**Code.** We release the full code for all our experiments at `https://github.com/kortukov/realistic_knowledge_conflicts`.

**Data and models.** We used the publicly available versions of all the datasets and models from the Huggingface hub. We used `chat` versions of the Llama models, `Instruct-v0.2` version of Mistral-7B, and `Instruct-v0.1` version of Mixtral-8x7B. We ran the Llama2-70B and Mixtral-8x7B models using 4-bit quantization, implemented in the `bitsandbytes` library. The 7B models were run without quantization.

**Hardware.** We ran experiments with 7B-sized models on a node with one Nvidia A100 GPU and two Intel Xeon Gold, 16 core, 2.9GHz CPUs, We ran the Llama2-70B and Mixtral-8x7B models on the same node using three Nvidia A100 GPUs, utilizing naive (vertical) model parallelism.

**Time.** The closed-book experiments on smaller models took 1-4 hours depending on the dataset. For the larger models, they took from 11 hours to 2 days and 20 hours. The open-book experiments took 35 minutes - 1.5 hours for the smaller models and 2 - 22 hours for the larger models.

# 9 Acknowledgements

The authors would like to thank Alexander Panfilov, Arnas Uselis, Bálint Mucsányi, and Michael Kirchhof for valuable insights and discussions.

This work was supported by the Tübingen AI Center. The authors thank the International Max Planck Research School for Intelligent Systems (IMPRS-IS) for supporting Alexander Rubinstein and Elisa Nguyen.

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

## A   Dataset sizes

For better understanding of the scale of our experiments, we report the size of the datasets at each stage of our experimental framework. Table 7 contains the results for each of the studied models.

As expected, the larger Llama2-70B and Mixtral-8x7B models have stronger closed-book performance than the smaller studied models. As a result, the knowledge conflict subset for these larger models are smaller. It is still large enough to draw meaningful statistical conclusions in our case. However, this trend illustrates one potential future limitation of our approach.

In this work, we rely on general knowledge QA datasets to study knowledge conflicts. As models grow bigger and memorize more and more facts from their training corpus, studying knowledge conflicts with our proposed realistic approach will require finding datasets that the models did not memorize. Examining the behavior of more powerful models under realistic knowledge conflicts will benefit from more long-tail and specialized question-answering datasets, on which these models do not perform well in a closed-book fashion.

(a) Llama2-7B model

| Dataset | Full data | Closed-book correct | Closed-book wrong | Knowledge conflict |
|---|---|---|---|---|
| NQ | 12,836 | 5,617 | 7,219 | 6,916 |
| SQuAD | 10,507 | 3,218 | 7,289 | 7,007 |
| NewsQA | 4,212 | 661 | 3,551 | 3,475 |
| TriviaQA | 7,785 | 5,169 | 2,616 | 2,555 |
| SearchQA | 16,980 | 12,333 | 4,647 | 4,575 |
| HotpotQA | 5,901 | 1,669 | 4,232 | 4,061 |

(b) Llama2-70B model

| Dataset | Full data | Closed-book correct | Closed-book wrong | Knowledge conflict |
|---|---|---|---|---|
| NQ | 12,836 | 7,326 | 5,510 | 5,161 |
| SQuAD | 10,507 | 4,081 | 6,426 | 6,127 |
| NewsQA | 4,212 | 845 | 3,367 | 3,251 |
| TriviaQA | 7,785 | 6,366 | 1,419 | 1,357 |
| SearchQA | 16,980 | 14,584 | 2,396 | 2,333 |
| HotpotQA | 5,901 | 2,292 | 3,609 | 3,401 |

(c) Mistral-7B model

| Dataset | Full data | Closed-book correct | Closed-book wrong | Knowledge conflict |
|---|---|---|---|---|
| NQ | 12,836 | 5,900 | 6,936 | 6,538 |
| SQuAD | 10,507 | 3,521 | 6,986 | 6,674 |
| NewsQA | 4,212 | 719 | 3,493 | 3,392 |
| TriviaQA | 7,785 | 5,593 | 2,192 | 2,119 |
| SearchQA | 16,980 | 12,882 | 4,098 | 4,019 |
| HotpotQA | 5,901 | 1,857 | 4,044 | 3,834 |

(d) Mixtral-8x7B model

| Dataset | Full data | Closed-book correct | Closed-book wrong | Knowledge conflict |
|---|---|---|---|---|
| NQ | 12,836 | 7,446 | 5,390 | 4,996 |
| SQuAD | 10,507 | 4,191 | 6,316 | 5,980 |
| NewsQA | 4,212 | 829 | 3,383 | 3,272 |
| TriviaQA | 7,785 | 6,535 | 1,250 | 1,185 |
| SearchQA | 16,980 | 14,469 | 2,511 | 2,435 |
| HotpotQA | 5,901 | 2,317 | 3,584 | 3,344 |

(e) GPT-3.5 Turbo model

| Dataset | Full data | Closed-book correct | Closed-book wrong | Knowledge conflict |
|---|---|---|---|---|
| NQ | 12,836 | 8,520 | 4,316 | 3,997 |
| SQuAD | 10,507 | 4,169 | 6,338 | 5,973 |
| NewsQA | 4,212 | 723 | 3,489 | 3,362 |
| TriviaQA | 7,785 | 6,947 | 838 | 766 |
| SearchQA | 16,980 | 15,219 | 1,761 | 1,695 |
| HotpotQA | 5,901 | 2,514 | 3,387 | 3,146 |

Table 7: Number of examples at each stage of the experimental pipeline for all studied models. The "Knowledge conflict" class contains examples that are input to **Stage 3** of experiments in which we measure knowledge update success.

# B  Prompt selection

In **Stage 3** of our experimental pipeline we evaluate the language model on an open-book QA task. When we execute experiments with prompting we rely on the zero-shot transfer capabilities of instruction-tuned language models. It is a well-known fact that such models are sensitive to the input prompts. To ensure the best possible performance on the task we search for the best-performing prompts on $3{,}000$ examples from the training split of each dataset. For each dataset, we choose the prompt with the highest open-book BEM score (percentage of correct answers according to BEM answer equivalence). Preliminary experiments have shown that our models are not sensitive to the format of the prompt. However, the performance varies significantly with different wordings of the instruction. We search over a space of 30 prompts that all have the same format:

```
prompt = f"""
Answer the question {instruction}.

Context: {context}
Question: {question}
Answer:
"""
```

The instruction consists of two parts that encode our knowledge of the answer format in the studied datasets.

```
instruction = f"{brevity_instruction} {context_reference}""
```

All possible options for each of the parts is listed in Table 8. The resulting optimal instruction wordings for each dataset are reported in Table 9.

| Aspect Varied | Examples |
|---|---|
| Brevity Instructions | - with as few words as possible
- concisely
- in the most compact form
- using minimal text
- by being succinct |
| Context References | - using the context
- by reading from the context
- by consulting the provided context
- through the information given in the context
- by extracting information directly from the context
- by copying verbatim from the context |

Table 8: All possible instruction wordings defining the search space for open-book QA prompt selection.

| Datasets | Optimal instruction wording |
|---|---|
| Natural Questions, SQuAD, NewsQA, HotpotQA | with as few words as possible by extracting information directly from the context |
| TriviaQA | with as few words as possible through the information given in the context |
| SearchQA | concisely using the context |

Table 9: Prompt selection results

## C   Motivation for studying the R subset

In § 4.3 and § 4.4 we focus on understanding the **R** subset of examples where the incorrect parametric answer is retained. In this section, we explain why we focus on this seemingly smaller subset of incorrect open-book answers.

When determining the importance of a problem, frequency should not be the sole factor. For example, adversarial attacks rarely happen in practice, yet there is substantial research to ensure the adversarial robustness of models. Analogously, the **R** subset may appear to be a fringe case, but it is a well-defined problem for which we provide a deeper analysis. Out of all incorrect open-book answers, the **R** subset contains only the ones that did not change at all after introducing the context. This specificity, together with our analysis is what makes this subset actionable.

The $\mathbf{U_i}$ subset, however, includes all other incorrect answer categories "lumped" together. If you look into the $\mathbf{U_i}$ subset, it is not one "prevalent scenario", but rather many of them. To illustrate this, we manually label 100 examples in the $\mathbf{U_i}$ subset for the Mixtral8x7B model on the Natural Questions dataset. We report the results in Table 10. As can be seen, analyzing the $\mathbf{U_i}$ subset would actually require separating each sub-category. This would turn each of them into actionable, but small subsets, similar to **R**.

| Number of examples | Type of error |
|---|---|
| 42 | Answer has an incorrect format. |
| 10 | Predicted answer is also correct, but not equal to the ground-truth answer. |
| 13 | Predicted answer is actually correct, while the dataset label is incorrect. |
| 12 | Answer is incorrect due to the model being bad at reading tables. |
| 13 | The predicted answer was part of the context, but not the correct answer. |
| 4 | Model predicts an answer that does not appear in context. |
| 5 | The model wrongly claims that the answer is not present in the context. |
| 3 | The context is illegible and or not enough to answer correctly. |

Table 10: Manually labeled reasons why examples are considered to be part of the $\mathbf{U_i}$ subset among 100 examples from Mixtral8x7B on Natural Questions.

# D  Parametric bias hypothesis testing

In our conflicted open-book QA experiments, all examples naturally split into two disjoint groups: in one the incorrect parametric answer $a_p$ appears in context $c$ (see § 4.3.1 for an example), and in the other group it does not. We report the sizes of the groups in Table 11. We model each of the groups of examples as a sequence of independent Bernoulli trials with an unknown success probability $\pi$. In our model success in the Bernoulli trial corresponds to a successful knowledge update, so $\pi$ is the theoretical quantity that the empirical probability $\mathbb{P}(\mathbf{U_c})$ estimates.

We ask the question: is there a statistically significant difference in the successful knowledge update probabilities in these two groups? We approach this problem from a Bayesian perspective and model the success probability $\pi_{\mathbf{U_c}}$ as a random variable. The number of successes in each group follows a Binomial distribution with $m$ successes out of $n$ trials:

$$p(m|n,\pi) = \binom{m}{n}\pi^m(1-\pi)^{n-m} \tag{1}$$

We choose the prior distribution on $\pi$ to be a uniform Beta distribution:

$$p(\pi) = \mathcal{B}(\pi; 1, 1) \tag{2}$$

The beta distribution is a conjugate prior to the Binomial likelihood and is the common choice when modeling success probabilities. The specific choice of parameters makes the prior uniform, encoding our lack of knowledge a priori. The posterior over $\pi$ after seeing $m$ successes in $n$ trials is then computable in closed-form and is also a Beta distribution:

$$p(\pi|m,n) = \mathcal{B}(\pi; m+1, n-m+1) \tag{3}$$

Let $\pi^0$ correspond to the success probability in $(a_p \not\subseteq c)$ group and $\pi^1$ is the success probability in $(a_p \subseteq c)$ group. Our null hypothesis is then: $\mathcal{H}_0 : \pi^0 = \pi^1$. This null hypothesis defines a predictive distribution $p(m_1|\mathcal{H}_0)$ over the observed number of successes in the group where $a_p \subseteq c$.

To test the null hypothesis we compute the probability of observing $m_1$ or less successes out of $n_1$ trials under the assumption that the null hypothesis is correct $p(m_1|\mathcal{H}_0)$. To compute the predictive distribution on $m_1$ we need to marginalize over the posterior on $\pi$:

$$p(m_1|\mathcal{H}_0) = \int p(m_1|n_1,\pi)p(\pi|m_0,n_0)d\pi \tag{4}$$

This results in a Beta-binomial distribution:

$$p(m_1|\mathcal{H}_0) = p(m_1|n_1, m_0, n_0) =$$
$$\binom{n_1}{m_1}\frac{\mathcal{B}(m_1+m_0+1, (n_1-m_1)+(n_0-m_0)+1)}{\mathcal{B}(m_0+1, (n_0-m_0)+1)} \tag{5}$$

We use this distribution to compute the final p-values by taking the value of the probability mass function at the observed value of $m_1$. Final results are reported in the main text in Table 4.

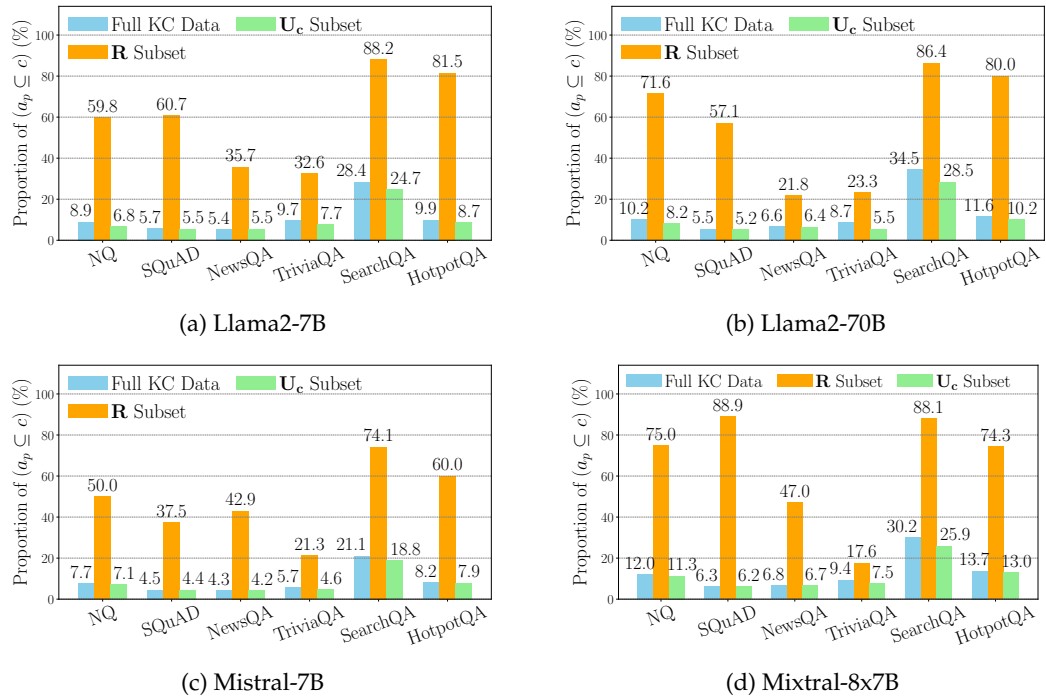

Figure 3: Frequency of samples that contain the incorrect parametric answer in the context $(a_p \subseteq c)$ in the full knowledge conflict (KC) data, the Retain subset (**R**) and the Correct update subset (**U$_c$**)for each studied LLM. Across all datasets and models, the percentage of context documents containing the incorrect parametric answer is largest in **R**.

|  | (a) Llama2-7B | |
|---|---|---|
| Dataset | Number of examples | |
|  | $(a_p \subseteq c)$ | $(a_p \not\subseteq c)$ |
| NQ | 615 | 6,303 |
| SQuAD | 403 | 6,604 |
| NewsQA | 188 | 3,287 |
| TriviaQA | 250 | 2,305 |
| SearchQA | 1,301 | 3,276 |
| HotpotQA | 401 | 3,661 |

|  | (b) Llama-70B | |
|---|---|---|
| Dataset | Number of examples | |
|  | $(a_p \subseteq c)$ | $(a_p \not\subseteq c)$ |
| NQ | 526 | 4,636 |
| SQuAD | 340 | 5,787 |
| NewsQA | 216 | 3,035 |
| TriviaQA | 118 | 1,239 |
| SearchQA | 804 | 1,529 |
| HotpotQA | 393 | 3,008 |

|  | (c) Mistral-7B | |
|---|---|---|
| Dataset | Number of examples | |
|  | $(a_p \subseteq c)$ | $(a_p \not\subseteq c)$ |
| NQ | 507 | 6,031 |
| SQuAD | 302 | 6,372 |
| NewsQA | 145 | 3,247 |
| TriviaQA | 120 | 1,999 |
| SearchQA | 847 | 3,171 |
| HotpotQA | 314 | 3,520 |

|  | (d) Mixtral-8x7B | |
|---|---|---|
| Dataset | Number of examples | |
|  | $(a_p \subseteq c)$ | $(a_p \not\subseteq c)$ |
| NQ | 602 | 4,390 |
| SQuAD | 375 | 5,605 |
| NewsQA | 224 | 3,048 |
| TriviaQA | 112 | 1,070 |
| SearchQA | 736 | 1,700 |
| HotpotQA | 459 | 2,890 |

Table 11: Number of examples where incorrect parametric answer $a_p$ appears in context $c$ for each dataset and model.

# E    On the importance of correct evaluation

In our three-stage experimental pipeline (§ 3.1), we prepare the data by filtering out examples that were correct closed-book or where the model answer did not conflict with the context. Unlike previous work, we focus on the closed-book incorrect examples. Thus, we need to identify incorrect model answers with high precision.

The commonly used Exact Match (EM) metric is prone to false negatives, as many conceptually equivalent answers may have different surface forms (and hence are considered different by EM). As pointed out by Bulian et al. (2022), EM only provides a lower bound on the system-level performance. However, EM is a weak example-level metric due to a large number of false negatives. Very often the model answers correctly but the surface form of the answer differs from the label (*Napoleon's* vs. *Napoleon*, *three* vs. *3* or *50–140 cm* vs. *0.5–1.4 m*). In our pipeline, using Exact Match would lead to a lot of actually correct (false negative) model answers making it into **Stage 3** of the pipeline. This would inflate the number of cases where the model retains its parametric answer, as there would be no reason for the model to update its answer since there is no conflict. For the same reason, we need to filter out incorrect but not conflicting examples in **Stage 2** of the pipeline using a separate NLI model that predicts whether the answer is entailed by the context.

Bulian et al. (2022) introduced BEM and showed it to be a better example-level metric than EM. We ported[1] the BEM metric to HF Transformers library and reproduced the original results. On the original data[2] BEM reaches 89% accuracy while EM only reaches 55 %. We further tested BEM on manually labeled subset of HotpotQA[3], where BEM resulted in 92% accuracy while EM only achieved 22%.

We perform an ablation study to illustrate the importance of BEM and NLI filtering by running the experiment on Llama2-7B without filtering and using the EM metric. We report the results in Table 12.

| Dataset | $\mathbb{P}(\mathbf{R})$ | $\mathbb{P}(\mathbf{U_c})$ | $\mathbb{P}(\mathbf{U_i})$ |
|---|---|---|---|
| NQ | 7.3 (+6.9) | 39.3 (−40.1) | 53.4 (+33.2) |
| SQuAD | 2.1 (+2.0) | 63.7 (−21.6) | 34.2 (+19.6) |
| NewsQA | 2.8 (+2.6) | 31.4 (−36.7) | 65.8 (+34.1) |
| TriviaQA | 9.2 (+5.9) | 55.5 (−23.1) | 35.4 (+17.4) |
| SearchQA | 8.9 (+8.2) | 28.8 (−31.1) | 62.2 (+22.8) |
| HotpotQA | 8.0 (+7.4) | 46.7 (−31.8) | 45.3 (+25.6) |

Table 12: Ablation study: knowledge update success rates for the Llama2-7B model when measured using Exact Match instead of BEM and not applying filtering. These results contain a lot of noise and thus are inflated. Size of change due to noise, compared to Table 3 in parenthesis.

Using a weak evaluation metric may lead one to wrongly conclude that the model often retains an incorrect parametric answer. This experiment highlights the importance of the strong answer equivalence metric (BEM, (Bulian et al., 2022)) and the filtering stage in our experimental pipeline. This design decision is crucial in identifying samples with a realistic knowledge conflict.

---

[1]BEM model available at: `https://huggingface.co/kortukov/answer-equivalence-bem`

[2]Original answer equivalence dataset: `https://huggingface.co/datasets/kortukov/answer-equivalence-dataset`

[3]HotpotQA answer equivalence dataset: `https://huggingface.co/datasets/kortukov/ae-gpt4-hotpotqa`

# F    Task adaptation using in-context learning

**Motivation**    In-context learning (ICL) is known to improve language model performance on a wide array of NLP tasks. ICL involves providing several labeled examples as part of the context. It is considered to be a method of adapting a general pre-trained model to perform a specific task. It is simpler and requires fewer resources than fine-tuning, but is generally more effective than instruction prompting alone.

**RQ**    Can in-context demonstrations minimize the influence of the discovered *parametric bias*?

**Evaluation setup**    The Llama2 model has a context length of 2048 tokens which does not allow for many in-context demonstrations. We therefore use five in-context examples for Natural Questions, SQuAD, and HotpotQA datasets, as they have shorter context lengths. For the other three datasets, we use two in-context demonstrations.

The Mistral-7B model employs sliding window attention, which enables flexible context lengths. Mixtral-8x7B has context length of 32768. This allows us to run the ICL experiment with eight context demonstrations for each dataset on both of those models.

**Results**    The results for the Llama2 family are presented in Table 13a and Table 13b. The 7B model continues to retain its parametric answer in roughly the same number of cases. Moreover, the incorrect parametric answer in the context still increases the probability of a knowledge update failure. For the larger 70B model, the trend is similar overall. Interestingly, for TriviaQA and SearchQA datasets Llama2-70B retains its parametric answer even more often with in-context demonstrations than without them. Furthermore, the negative influence of the parametric answer on update probability exists for all datasets. We conclude that for the Llama2 model family, in-context demonstrations do not minimize the influence of the *parametric bias*.

The results for Mistral-7B and Mixtral-8x7B are reported in Table 13c and Table 13d. The influence of in-context demonstrations is two-fold: it improves open-book performance ($\mathbb{P}(\mathbf{U_c})$), but also makes the model retain the incorrect parametric answer more often than without ICL. Furthermore, the parametric bias (difference in failure likelihood) is slightly exacerbated after introducing ICL for the majority of datasets.

**Limitations**    Due to a limited context window, shown examples might not suffice for the model to learn the task. This phenomenon was explored in the literature for text classification tasks by the name of "demonstration bias" (Fan et al., 2024). This could explain the marginal improvement in the performance and ICL failing to prevent parametric bias.

**Conclusion**    For all studied models better task adaptation through in-context learning ***does not mitigate the discovered parametric bias.***

(a) Llama2-7B

| Dataset | $\mathbb{P}(\mathbf{R})$ | $\mathbb{P}(\mathbf{U_c})$ | $\mathbb{P}(\mathbf{U_i})$ | $\Delta\mathbb{P}(\mathbf{R})$ |
|---|---|---|---|---|
| NQ | 1.8 | 82.0 | 16.2 | **+11.4*** |
| SQuAD | 0.3 | 92.6 | 7.1 | **+2.7*** |
| NewsQA | 0.3 | 73.0 | 26.6 | **+2.0*** |
| TriviaQA | 3.8 | 79.8 | 16.3 | **+8.9*** |
| SearchQA | 3.8 | 55.6 | 40.6 | **+10.5*** |
| HotpotQA | 1.2 | 84.1 | 14.7 | **+7.3*** |

(b) Llama2-70B

| Dataset | $\mathbb{P}(\mathbf{R})$ | $\mathbb{P}(\mathbf{U_c})$ | $\mathbb{P}(\mathbf{U_i})$ | $\Delta\mathbb{P}(\mathbf{R})$ |
|---|---|---|---|---|
| NQ | 2.0 | 84.2 | 13.7 | **+13.2*** |
| SQuAD | 0.2 | 95.6 | 4.2 | **+2.2*** |
| NewsQA | 0.8 | 75.6 | 23.6 | **+4.7*** |
| TriviaQA | 6.1 | 76.5 | 17.4 | **+11.5*** |
| SearchQA | 8.9 | 70.1 | 20.9 | **+20.8*** |
| HotpotQA | 1.0 | 89.7 | 9.3 | **+5.8*** |

(c) Mistral-7B

| Dataset | $\mathbb{P}(\mathbf{R})$ | $\mathbb{P}(\mathbf{U_c})$ | $\mathbb{P}(\mathbf{U_i})$ | $\Delta\mathbb{P}(\mathbf{R})$ |
|---|---|---|---|---|
| NQ | 1.0 | 85.8 | 13.1 | **+8.5*** |
| SQuAD | 0.1 | 93.8 | 6.0 | **+2.7*** |
| NewsQA | 0.1 | 74.5 | 25.3 | **+4.5*** |
| TriviaQA | 4.1 | 81.0 | 14.8 | **+12.9*** |
| SearchQA | 3.8 | 74.0 | 22.1 | **+8.9*** |
| HotpotQA | 0.8 | 86.2 | 12.9 | **+5.7*** |

(d) Mixtral-8x7B

| Dataset | $\mathbb{P}(\mathbf{R})$ | $\mathbb{P}(\mathbf{U_c})$ | $\mathbb{P}(\mathbf{U_i})$ | $\Delta\mathbb{P}(\mathbf{R})$ |
|---|---|---|---|---|
| NQ | 2.7 | 84.8 | 12.4 | **+11.1*** |
| SQuAD | 0.1 | 95.5 | 4.3 | **+0.5*** |
| NewsQA | 0.8 | 74.8 | 24.3 | **+3.5*** |
| TriviaQA | 8.5 | 75.5 | 15.9 | **+9.6*** |
| SearchQA | 7.4 | 70.7 | 21.9 | **+18.2*** |
| HotpotQA | 1.2 | 86.6 | 12.1 | **+5.2*** |

Table 13: Knowledge update success rates of each model when shown in-context demonstrations. We report empirical probabilities (in %) for each model that it retains the parametric answer ($\mathbb{P}(\mathbf{R})$), successfully updates its answer to the correct contextual one ($\mathbb{P}(\mathbf{U_c})$), or updates to an incorrect answer ($\mathbb{P}(\mathbf{U_i})$). We additionally report the difference in the likelihood (in %) of knowledge update failure ($\Delta\mathbb{P}(\mathbf{R})$) for ($a_p \subseteq c$) versus ($a_p \nsubseteq c$). A positive difference means the parametric answer in context increases failure likelihood. Asterisk * denotes $p < 0.05$.

# G    Task adaptation using fine-tuning

**Motivation**    Fine-tuning a pre-trained language model on a downstream task is a powerful task adaptation method. Fine-tuning involves further training an LLM on a set of labeled examples. It requires access to model weights and more computational resources than ICL or prompting but generally enables stronger downstream task performance. One caveat of fine-tuning is that by changing the model weights we may lose the generality of an instruction-tuned model.

**RQ**    Can stronger task adaptation with fine-tuning minimize the influence of the discovered *parametric bias*?

**Evaluation setup**    We fine-tune the Llama2-7B model on the training subsets of the studied datasets. For example, the model evaluated on the validation subset of NQ was only trained on the training subset of NQ. We fine-tune the 4-bit quantized model using QLoRA applied to all linear layers. On all datasets, we train for 1 epoch with batch size 2, a constant learning rate of 2e-4, LoRA dropout set to 0.05 and not updating the biases. The dataset-specific (tuned) hyperparameters are reported in Table 14.

| Dataset | # train examples | LoRA r | LoRA alpha |
|---|---|---|---|
| NQ | 104,071 | 4 | 2 |
| SQuAD | 86,588 | 4 | 2 |
| NewsQA | 10,000 | 8 | 8 |
| TriviaQA | 10,000 | 8 | 8 |
| SearchQA | 10,000 | 32 | 64 |
| HotpotQA | 72,928 | 16 | 32 |

Table 14: Hyperparameters in the fine-tuning experiment.

**Results**    We compare the fine-tuned model with the original prompted version used in the main experiments.

First, we report our observations on knowledge-updating behaviors and the influence of an incorrect parametric answer naturally appearing in the context in Table 15. As expected, fine-tuning improves open-book accuracy across all studied datasets. We find that fine-tuning reduces the chance of the model retaining its original incorrect answer. The fine-tuned version retains the original answer in only 1.2% of cases on average across datasets, compared to 1.6% for the prompted model. Regarding the *parametric bias*, we find that when an incorrect parametric answer naturally appears in context - its influence on the failure likelihood is smaller for the fine-tuned model across all datasets.

Additionally, we run an intervention study, to compare how the fine-tuned model responds to artificially adding the incorrect parametric answers to context. We follow the setup in § 4.4.2.

The results are reported in Table 16. The fine-tuned model is still susceptible to adding the incorrect parametric answer in context, however, to a lesser extent than the prompted version. For the original model, the likelihood of retaining the incorrect answer increases by 7.9% on average across all datasets. For the fine-tuned model the average increase only reaches 4.3%.

**Discussion**    Fine-tuning reduces the *parametric bias* in our experiments. However, in the examined setup fine-tuning was not enough to completely eradicate it. Overall, our results suggest that fine-tuning is the most promising approach to combat the parametric bias in RAG systems. However, it comes at the cost of potentially losing the generality of an instruction-tuned LLM.

(a) Llama2-7B
Prompted

| Dataset | $\mathbb{P}(\mathbf{R})$ | $\mathbb{P}(\mathbf{U_c})$ | $\mathbb{P}(\mathbf{U_i})$ | $\Delta\mathbb{P}(\mathbf{R})$ |
|---|---|---|---|---|
| NQ | 1.4 | 79.6 | 19.0 | **+8.8\*** |
| SQuAD | 0.4 | 90.3 | 9.3 | **+4.0\*** |
| NewsQA | 0.8 | 72.0 | 27.1 | **+4.8\*** |
| TriviaQA | 3.4 | 79.3 | 17.3 | **+8.7\*** |
| SearchQA | 2.3 | 61.5 | 36.3 | **+6.6\*** |
| HotpotQA | 1.3 | 79.6 | 19.0 | **+10.7\*** |

(b) Llama2-7B
Fine-tuned

| Dataset | $\mathbb{P}(\mathbf{R})$ | $\mathbb{P}(\mathbf{U_c})$ | $\mathbb{P}(\mathbf{U_i})$ | $\Delta\mathbb{P}(\mathbf{R})$ |
|---|---|---|---|---|
| NQ | 0.2 | 89.0 | 10.9 | **+0.9\*** |
| SQuAD | 0.0 | 95.5 | 4.4 | **+0.6** |
| NewsQA | 0.3 | 75.5 | 24.2 | **+2.5\*** |
| TriviaQA | 4.4 | 82.1 | 13.6 | **+5.7\*** |
| SearchQA | 1.7 | 85.3 | 13.0 | **+2.6\*** |
| HotpotQA | 0.4 | 90.7 | 8.9 | **+1.4\*** |

Table 15: Knowledge update success rates of Llama2-7B prompted and fine-tuned. We report empirical probabilities (in %) for each model that it **R**etains the parametric answer ($\mathbb{P}(\mathbf{R})$), successfully updates its answer to the correct contextual one ($\mathbb{P}(\mathbf{U_c})$), or updates to an incorrect answer ($\mathbb{P}(\mathbf{U_i})$). We additionally report the difference in the likelihood (in %) of knowledge update failure (**R**) for ($a_p \subseteq c$) versus ($a_p \not\subseteq c$). A positive difference means the parametric answer in context increases failure likelihood. Asterisk * denotes $p < 0.05$.

| Dataset | Llama2-7B Prompted | | Llama2-7B Fine-tuned | |
|---|---|---|---|---|
| | $\mathbb{P}(\mathbf{R})$ | $\mathbb{P}(\mathbf{U_c})$ | $\mathbb{P}(\mathbf{R})$ | $\mathbb{P}(\mathbf{U_c})$ |
| NQ | 12.5 (+11.1) | 70.1 (−9.5) | 3.9 (+3.7) | 85.9 (−3.1) |
| SQuAD | 12.0 (+11.6) | 77.3 (−0.4) | 2.7 (+2.7) | 92.7 (−2.8) |
| NewsQA | 7.6 (+6.8) | 66.2 (−5.8) | 4.0 (+3.7) | 72.7 (−2.8) |
| TriviaQA | 11.0 (+7.6) | 76.2 (−3.1) | 14.1 (+9.7) | 77.1 (−5.0) |
| SearchQA | 6.6 (+4.3) | 60.5 (−1.0) | 2.7 (+1.0) | 83.2 (−2.1) |
| HotpotQA | 7.3 (+6.0) | 72.8 (−6.8) | 5.4 (+5.0) | 86.3 (−4.4) |

Table 16: Knowledge update success rate (%) after **adding** the parametric answer $a_p$ to the context after words *"Unrelated text:"* (details in § 4.4.2). Size of change in parenthesis. We compare the prompted and fine-tuned versions of Llama2-7B.

## H   Influence of model size on the parametric bias

**Motivation**   Given enough training data, larger LLMs tend to perform better on both the pretraining objective and downstream tasks.

**RQ**   How does the size of a language model influence its knowledge-updating behavior and susceptibility to *parametric bias*?

**Evaluation setup**   We compare two models of the same architecture that differ only in parameter count: Llama2-7B and Llama2-70B.

**Results**   First, we report our observations on knowledge updating behaviors and the influence of an incorrect parametric answer naturally appearing in the context in Table 17. We find that 7B and 70B models have very similar behaviors with respect to knowledge updating: Llama2-7B retains its original incorrect answer in only 1.6% of examples on average and Llama2-70B does that on average for 2% of examples. We also find that the two models are similarly susceptible to failing their knowledge updates when their incorrect parametric answer appears in context.

Additionally, we run an intervention study, to compare how models of different sizes respond to artificially adding their incorrect parametric answers to context. We follow the setup in § 4.4.2.

The results are reported in Table 18. Both models are susceptible to parametric bias: adding the incorrect parametric answer to the context increases their likelihood of retaining it even after seeing the context. For Llama2-7B the increase is on average by 7.9 %pt. For the larger 70B model, this increase is on average by 4.8 %pt.

**Discussion**   These experiments provide evidence that ***increasing the model size can be a viable, although costly, solution to mitigate the discovered parametric bias.*** However, among the currently popular open models, ***even the largest ones are susceptible to this bias***.

(a) Llama2-7B

| Dataset | $\mathbb{P}(\mathbf{R})$ | $\mathbb{P}(\mathbf{U_c})$ | $\mathbb{P}(\mathbf{U_i})$ | $\Delta\mathbb{P}(\mathbf{R})$ |
|---------|------|------|------|--------|
| NQ | 1.4 | 79.6 | 19.0 | **+8.8\*** |
| SQuAD | 0.4 | 90.3 | 9.3 | **+4.0\*** |
| NewsQA | 0.8 | 72.0 | 27.1 | **+4.8\*** |
| TriviaQA | 3.4 | 79.3 | 17.3 | **+8.7\*** |
| SearchQA | 2.3 | 61.5 | 36.3 | **+6.6\*** |
| HotpotQA | 1.3 | 79.6 | 19.0 | **+10.7\*** |

(b) Llama2-70B

| Dataset | $\mathbb{P}(\mathbf{R})$ | $\mathbb{P}(\mathbf{U_c})$ | $\mathbb{P}(\mathbf{U_i})$ | $\Delta\mathbb{P}(\mathbf{R})$ |
|---------|------|------|------|--------|
| NQ | 2.0 | 84.2 | 13.7 | **+11.4\*** |
| SQuAD | 0.2 | 93.3 | 6.5 | **+2.3\*** |
| NewsQA | 1.0 | 72.9 | 26.0 | **+2.4\*** |
| TriviaQA | 4.0 | 65.3 | 30.7 | **+7.6\*** |
| SearchQA | 4.1 | 65.8 | 30.0 | **+9.5\*** |
| HotpotQA | 1.0 | 87.0 | 12.0 | **+6.9\*** |

Table 17: Knowledge update success rates of two models of Llama2 family of 7B and 70B sizes. We report empirical probabilities (in %) for each model that it **R**etains the parametric answer ($\mathbb{P}(\mathbf{R})$), successfully updates its answer to the correct contextual one ($\mathbb{P}(\mathbf{U_c})$), or updates to an incorrect answer ($\mathbb{P}(\mathbf{U_i})$). We additionally report the difference in the likelihood (in %) of knowledge update failure (**R**) for ($a_p \subseteq c$) versus ($a_p \not\subseteq c$). A positive difference means the parametric answer in context increases failure likelihood. Asterisk * denotes $p < 0.05$.

| | Llama2-7B | | Llama2-70B | |
|---------|------|------|------|------|
| Dataset | $\mathbb{P}(\mathbf{R})$ | $\mathbb{P}(\mathbf{U_c})$ | $\mathbb{P}(\mathbf{R})$ | $\mathbb{P}(\mathbf{U_c})$ |
| NQ | 12.5 (+11.1) | 70.1 (−9.5) | 7.9 (+6.2) | 72.2 (−7.9) |
| SQuAD | 12.0 (+11.6) | 77.3 (−0.4) | 3.5 (+3.3) | 88.0 (−5.3) |
| NewsQA | 7.6 (+6.8) | 66.2 (−5.8) | 6.9 (+5.9) | 68.3 (−4.6) |
| TriviaQA | 11.0 (+7.6) | 76.2 (−3.1) | 11.6 (+7.6) | 63.6 (−1.7) |
| SearchQA | 6.6 (+4.3) | 60.5 (−1.0) | 8.0 (+3.9) | 61.5 (−4.3) |
| HotpotQA | 7.3 (+6.0) | 72.8 (−6.8) | 3.2 (+2.2) | 84.4 (−2.6) |

Table 18: Knowledge update success rate (%) after **adding** the parametric answer $a_p$ to the context after words *"Unrelated text:"* (details in § 4.4.2). Size of change in parenthesis. We compare 7B and 70B versions of the Llama2 model.

# I   Parametric answer is likely to appear in real-world documents

Retrieval-augmented generation is designed to address scenarios where the factual information is changing. Supplying up to date documents (e.g. web pages) as context for LLM inference then allows to update the system's knowledge and provide correct information to the end user. We argue, that in these scenarios the parametric answer of the model is likely to appear in the retrieved document.

First, the obsolete parametric answer is not random and still very related to the question and the document that is supposed to answer it. Even if the answer is currently incorrect, it is likely to be a previously correct answer, or the model's "best guess" at answering the question. This is especially true for larger models that store a lot of factual information in their parameters after the pre-training stage. Second, large and unstructured documents, especially web pages, may contain background information and historical data together with the current information. For example, the Wikipedia article for the President of the United States contains the text *"Donald Trump"* seven times, as of this writing.

To ground this argument in a quantitative estimate, we employ the FreshQA benchmark (Vu et al., 2023). It is a weekly updated QA dataset containing factual questions whose answers change with time. FreshQA contains questions together with answers and a URL for a source web page containing the answer. We ask the question: *How often does a parametric answer of a language model appear in the source web page?*

We only consider the subset of FreshQA where the answers change over time (`slow-changing` and `fast-changing` examples). When searching for the parametric answer to appear on the web page we use the jusText python library to extract text.

We report the results in Table 19. As we can see, the parametric answer of the studied LLMs is 23-33% likely to appear on the source web page. Moreover, a significant portion of those answers are wrong, indicating that the parametric knowledge of the model is obsolete in those cases.

Real-world large retrieved documents can often contain the parametric answer of a language model. As we show in this paper, this appearance can influence the knowledge-updating behavior of the model. These results highlight the importance and relevance of the found *parametric bias* phenomenon for real-world retrieval-augmented systems.

| Model | Parametric answer in web page | Incorrect |
|---|---|---|
| Llama2-7B | 33.5% | 62.1% |
| Llama2-70B | 32.7% | 57.8% |
| Mistral-7B | 23.2% | 44.2% |
| Mixtral-8x7B | 29.0% | 54.2% |

Table 19: Frequency of the parametric answer appearing on the source web page and which portion of those answers is incorrect. We report the results on the FreshQA dataset.

# J GPT results

**Motivation**   Closed-source LLMs are widely used in practice for RAG-based applications, despite being less promising for research due to their API-only access. We include one such model for the comprehensiveness of our study.

**RQ**   Do our findings generalize to the GPT family of models?

**Setup**   We run the OpenAI GPT-3.5-Turbo-0125 using the experimental setup presented in the main text. The dataset sizes at every step of the experimental pipeline can be found in Table 7.

**Knowledge updating behaviors**   This experiment follows § 4.2. We report results in Table 20. GPT-3.5 retains its original parametric answer on average in 3.5% of cases. Similar to the studied open-source models, GPT-3.5 tends to update its knowledge from the context when presented with real conflicting documents.

| | GPT-3.5-Turbo | | |
|---|---|---|---|
| Dataset | $\mathbb{P}(\mathbf{R})$ | $\mathbb{P}(\mathbf{U_c})$ | $\mathbb{P}(\mathbf{U_i})$ |
| NQ | 1.7 | 84.0 | 14.3 |
| SQuAD | 0.1 | 95.5 | 4.3 |
| NewsQA | 0.5 | 77.5 | 22.0 |
| TriviaQA | 12.0 | 67.1 | 20.9 |
| SearchQA | 5.4 | 72.2 | 22.4 |
| HotpotQA | 1.3 | 87.0 | 11.7 |
| Average | 3.5 | 80.6 | 15.9 |

Table 20: Categorization of conflicted open-book QA answers of GPT-3.5-Turbo model. We report the proportion of open-book answers (in %) where the model retains its parametric answer ($\mathbb{P}(\mathbf{R})$), successfully updates its answer to the correct contextual one ($\mathbb{P}(\mathbf{U_c})$), or updates its answer incorrectly ($\mathbb{P}(\mathbf{U_i})$), as defined in § 3.2.

**Parametric bias influence**   We test for the influence of parametric bias as in § 4.3.2. The results are reported in Table 21. For all six studied datasets, the incorrect parametric answer in context increases the likelihood of a failed knowledge update.

| Dataset | GPT-3.5 Turbo |
|---|---|
| NQ | **+13.0*** |
| SQuAD | **+2.4*** |
| NewsQA | **+7.6*** |
| TriviaQA | **+21.4*** |
| SearchQA | **+11.0*** |
| HotpotQA | **+6.9*** |

Table 21: Difference in the likelihood (in %) of knowledge update failure (**R**) for $(a_p \subseteq c)$ versus $(a_p \not\subseteq c)$. A positive difference means the parametric answer in context increases failure likelihood. Asterisk * denotes statistical significance at $\alpha = 0.05$.

**Masking the parametric answer**   This section follows the setup of § 4.4.1. We report the results in Table 22. For the GPT-3.5 model masking the incorrect parametric answer if it appears in the context reduces the likelihood that the model retains its original answer.

**Artificially adding the parametric answer**   In this experiment, we add the original parametric answer to the context preceded by the phrase "Unrelated text: ", as in § 4.4.2. The results can be found in Table 23. Similar to the open-source models, GPT-3.5 Turbo retains its original answer more often after this intervention. This holds for all studied datasets.

| | GPT-3.5 Turbo | |
|---|---|---|
| Dataset | $\mathbb{P}(\mathbf{R})$ | $\mathbb{P}(\mathbf{U_c})$ |
| NQ | 0.7 (−1.0) | 84.4 (−0.4) |
| SQuAD | 0.1 ( 0.0) | 95.4 (−0.1) |
| NewsQA | 0.3 (−0.2) | 77.7 (+0.2) |
| TriviaQA | 9.1 (−2.9) | 68.0 (+0.9) |
| SearchQA | 3.3 (−2.1) | 72.0 (−0.2) |
| HotpotQA | 0.7 (−0.6) | 87.5 (−0.5) |

Table 22: Knowledge update success rate (%) after **masking** the parametric answer $a_p$ if it appears in the context. Size of change in parenthesis.

| | GPT-3.5 Turbo | |
|---|---|---|
| Dataset | $\mathbb{P}(\mathbf{R})$ | $\mathbb{P}(\mathbf{U_c})$ |
| NQ | 3.3 (+1.6) | 75.1 (−8.9) |
| SQuAD | 0.5 (+0.4) | 94.3 (−1.2) |
| NewsQA | 1.1 (+0.6) | 76.2 (−1.3) |
| TriviaQA | 21.5 (+9.5) | 62.0 (−5.1) |
| SearchQA | 8.0 (+2.6) | 66.1 (−6.1) |
| HotpotQA | 2.4 (+1.1) | 84.9 (−2.1) |

Table 23: Knowledge update success rate (%) after **adding** the parametric answer $a_p$ to the context after words *"Unrelated text:"*. Size of change in parenthesis.

**Conclusion**   In our experiments, GPT-3.5-Turbo exhibits the same behaviors as the studied open-source models. It readily updates its knowledge from real documents and it is also susceptible to the discovered parametric bias.

