# OpenReview forum: "Studying Large Language Model Behaviors Under Context-Memory Conflicts With Real Documents"
_colmweb.org/COLM/2024/Conference — COLM_

### Official Review · Reviewer_y1YK · 2024-04-27

**Rating:** 6
**Confidence:** 5
**Ethics Flag:** 1

**Summary:**

This paper proposes a framework to investigate the knowledge conflict problem in the RAG scenarios. Compared with previous work, the authors claim that they use real conflicting documents rather than synthetic information, which is closer to the real-world setting. According to this paper, LLMs are easy to update their parametric memory when external conflicting evidence is presented in the context. Furthermore, the authors find a 'parametric bias' phenomenon: It's hard for LLMs to update their parametric memory if there is parametric memory-related information in the context, although it is incorrect and unrelated to the question.

**Questions To Authors:**

1. Could the phenomenon of 'masking reducing the likelihood of retaining parametric answers' be attributed to low-quality masking? It's possible that LLMs are strong enough to identify such obvious misinformation?

**Reasons To Accept:**

1. The knowledge conflict is an important problem in the RAG scenario, and the authors' findings are really helpful for the community.
2. This paper is easy to follow.
3. The experiments are sufficient and the analysis is promising.

**Reasons To Reject:**

1. In stage 1, the authors mention using in-context demonstration to elicit answers. However, could this introduce demonstration bias? The authors should clarify this aspect in the paper.
2. In stage 2, the authors mention using BEM to match the answers. What is the accuracy of BEM? Can it ensure matching quality? The authors should provide clarification on this in the paper.
3. Given the widespread use of closed-source LLMs in real-world RAG scenarios, it's essential to include at least one closed-source LLM to ensure the findings can be generalized to these models.
4. Why do the authors mention "tune the prompts" in Stage 3? I've checked the appendix, but I'm still confused about this point. Does it mean the authors have fine-tuned the models, or have they simply selected one of the best prompts? If it's the former, could this introduce another training bias since the parameters have been changed? If it's the latter, the authors should reconsider the terminology.
5. The first finding doesn't seem entirely new, as it has been reported in "Adaptive Chameleon or Stubborn Sloth: Revealing the Behavior of Large Language Models in Knowledge Conflicts," which is also referenced in this paper. Specifically, Xie et al. found that LLMs are highly likely to update their parametric memory as long as the external evidence is coherent. Could you please justify the difference?

(I will reconsider the score if the authors can address the points mentioned above to my satisfaction.)

---

> ### Author Rebuttal · Authors · 2024-05-30
>
> We thank the reviewer for constructive feedback and questions. We answer them below:
> > 1.using in-context demonstration to elicit answers. However, could this introduce demonstration bias?
>
> We understand demonstration bias for generative QA as “shown examples do not suffice to learn the task” (analogous to [a] for classification). Due to the limited context window, it could explain marginal improvement in the performance and ICL failing to prevent parametric bias (App. E). We’ll add this discussion to the revision. If we misunderstood, we kindly ask the reviewer to be more specific.
>
> [a] https://arxiv.org/pdf/2312.07476
> > 2.What is the accuracy of BEM? Can it ensure matching quality?
>
> [b] showed BEM to be a stronger answer matching metric than EM. In our experiments it also outperforms EM (89% vs. 55% on original [b] data, 92% vs. 22% on our manually labeled test data from HotpotQA). We’ll add this to App. D.
>
> [b] https://arxiv.org/abs/2202.07654
> > 3.it's essential to include at least one closed-source LLM.
>
> We will add results for GPT-3.5 (below).
>
> Table 3, Conflicted open-book QA
> |Dataset|R|Uc|Ui|
> |-|-|-|-|
> |NQ|1.7|84.0|14.3|
> |SQuAD|0.1|95.5|4.3|
> |NewsQA|0.5|77.5|22.0|
> |TriviaQA|12.0|67.1|20.9|
> |SearchQA|5.4|72.2|22.4|
> |HotpotQA|1.3|87.0|11.7|
>
> Table 4 Diff. in the likelihood of update failure
> |Dataset|GPT-3.5|
> |-|-|
> |NQ|+10.3*|
> |SQuAD|+0.9|
> |NewsQA|+5.9|
> |TriviaQA|+17.2*|
> |SearchQA|+16.2*|
> |HotpotQA|+6.8*|
>
> > 4. Does it mean the authors have fine-tuned the models, or have they simply selected one of the best prompts?
>
> We selected the best prompt. We did not fine-tune the models. We will clarify this in the revision.
> > 5. Xie et al. found that LLMs are highly likely to update their parametric memory as long as the external evidence is coherent. Could you please justify the difference?
>
> Xie et al. study coherent yet false contexts. In our setup we check behaviors for real (both coherent and factually true) contexts, which allowed us to find the parametric bias.
>
> > Q1. Could the phenomenon of 'masking reducing the likelihood of retaining parametric answers' be attributed to low-quality masking?
>
> In our understanding, you suggest masking makes inputs OOD (“missingness bias” [c]). To minimize it, we test various strategies: using attention mask [c], varying mask tokens, using paraphrases, and changing mask size. We use the strategy with the best open-book accuracy. We ask the reviewer to elaborate if we misunderstood.
>
> [c] https://arxiv.org/abs/2204.08945

---

> > ### Comment · Reviewer_y1YK · 2024-06-04
> > **Reply to the Authors**
> >
> > Sincerely thank you for the detailed response. I have adjusted my score accordingly. I encourage the authors to incorporate these discussions into the next version. One last comment: If I remember correctly, Xie et al. studied both correct and false contexts. Perhaps the authors should verify this and include the differences in the next version.

---

> > > ### Author Response · Authors · 2024-06-04
> > >
> > > Thank you for considering the rebuttal and changing your score. We will improve the revision and include the discussed points.
> > > We also re-checked Xie et al.'s work and would like to elaborate on the difference between ours and their work:
> > >
> > > We refer to single-source experiments (§4.1 in Xie et al.) when we say they studied coherent yet false contexts.
> > > In these experiments, Xie et al. provide a "counter-memory" context which they synthetically generate to contradict the model's knowledge.
> > > Our setting is also single-source, but the contexts are real and factually true, hence we claim our setting is more realistic.
> > >
> > > Xie et al. also have multi-source evidence experiments (§4.2).
> > > These experiments contain, as you rightly point out, "both correct and false contexts".
> > > Their multi-source experiments focus on "the evidence preference of LLMs", while we focus on knowledge-updating behavior.
> > > Their experiments show LLMs exhibit a *confirmation bias* - they often prefer evidence supporting their parametric knowledge.
> > >
> > > Our realistic single-source experiments extend their findings. We show that real documents meant to update knowledge can unintentionally bias the model toward its wrong parametric answer (our section 4.3.1). This can be framed as a confirmation bias of sorts, which we term "parametric bias". The parametric bias describes the case when models fail to update their knowledge even if the evidence is **not** supporting the parametric knowledge but just includes the parametric answer.
> > > We will add this distinction to the paper.

---

> > > > ### Comment · Reviewer_y1YK · 2024-06-04
> > > > **Reply to the Authors (2)**
> > > >
> > > > Thank you for your prompt and detailed explanation. I have no further questions. Looking forward to the revision.

---

### Official Review · Reviewer_gWDt · 2024-05-03

**Rating:** 6
**Confidence:** 3
**Ethics Flag:** 1

**Summary:**

This paper studies knowledge conflicts under a new setup: incorrect parametric knowledge conflicts with ground-truth knowledge in context. The paper discovers parametric bias, which means knowledge
updates can be impeded if incorrect parametric answers appear in the retrieved documents. Extensive experiments are conducted to demonstrate this.

**Questions To Authors:**

1. Why the happening of knowledge updates is determined by exact match? Will changing it into semantic equality change the results much?

**Reasons To Accept:**

1. Probing the incorrect parametric knowledge to create knowledge conflicts is novel and interesting.
2. The experiments are comprehensive and well-designed. They fully demonstrate the main conclusion.
3. The paper is well-written and easy to follow.

**Reasons To Reject:**

1. While the setup differs from previous works, the main finding seems not very novel.
2. The paper does not provide a solution. It mainly focuses on analyzing and demonstrating a phenomenon.
3. The paper emphasizes knowledge conflicts under its setup are realistic. However, in reality, the retrieved knowledge might be noisy and not necessarily be the ground truth.

---

> ### Author Rebuttal · Authors · 2024-05-30
>
> We thank the reviewer for recognizing the merits of our work and providing constructive feedback. We address concerns in the following:
> > While the setup differs from previous works, the main finding seems not very novel.
>
> Xie et al. report a similar finding for coherent yet false contexts. In our setup, we check behaviors in case of real (coherent **and** factually true) contexts, allowing us to find the parametric bias.
> >The paper does not provide a solution. It mainly focuses on analyzing and demonstrating a phenomenon.
>
> Indeed, we discover and examine the problem and its scope, and invite the community to work together on solving it. To facilitate, we try simple solutions:
> - ICL task adaptation (App. E)
> - Increasing model size (App. F)
> - Fine-tuning - our best result (below, will be added to revision).
>
> We fine-tune Llama7B on the train split of each dataset.
>
> (R) subset becomes smaller, but the model is still susceptible to adding the parametric answer to context.
>
> Open-book QA under knowledge conflict (cf. Section 4.2)
> |Data| R|Uc|Ui|
> |-|-|-|-|
> |NQ|0.1|89.0|10.9|
> |SQuAD|0.0|95.5|4.4|
> |NewsQA|0.3|75.5|24.3|
> |TriviaQA|4.4|82.1|13.6|
> |SearchQA|1.7|85.3|13.0|
> |HotpotQA|0.4|90.7|8.9|
>
> After adding the incorrect parametric answer (cf. section 4.4.2):
> |Dataset|R|Uc|Ui|
> |-|-|-|-|
> |NQ|3.9|85.9|10.2|
> |SQuAD|2.7|92.7|4.6|
> |NewsQA|4.0|72.7|23.3|
> |TriviaQA|14.0|77.1|8.8|
> |SearchQA|2.7|82.2|14.0|
> |HotpotQA|5.4|86.3|8.4|
>
> > in reality, the retrieved knowledge might be noisy and not necessarily be the ground truth.
>
> Noisy retrieved documents are indeed realistic. However, prior work studied **synthetic documents designed to contradict** model knowledge, which is unrealistic. We study real documents, which is more realistic.
> > Why the happening of knowledge updates is determined by exact match?
>
> 1. To be comparable to previous work
> 2. RAG is designed to update model knowledge. If context had 0 influence on the answer, it is a critical failure of RAG.
>
> > Will changing it into semantic equality change the results much?
>
> We re-ran experiments on Llama7B using BEM to determine knowledge update.
>
> Table 3, open-book QA on the KC subset
> |Dataset|R|Uc|Ui|
> |-|-|-|-|
> |NQ|4.8|79.9|15.3|
> |SQuAD|1.6|90.3|8.1|
> |NewsQA|4.9|72.3|22.8|
> |TriviaQA|5.7|79.5|14.7|
> |SearchQA|7.3|62.0|30.7|
> |HotpotQA|2.7|79.6|17.8|
>
> The general trends are the same, but more examples are in R as more answer pairs are equivalent according to BEM. Please let us know if there are any further questions.

---

> > ### Comment · Reviewer_gWDt · 2024-06-05
> >
> > Thank you for the detailed response. I have no further questions. I will keep the score of 6.

---

### Official Review · Reviewer_2iuH · 2024-05-10

**Rating:** 5
**Confidence:** 4
**Ethics Flag:** 1

**Summary:**

This work proposes to study another scenario of LLM knowledge conflicts: when the LLM parametric knowledge is wrong and the retrieved document contains evidence that supports the correct answer. Experiments demonstrate that models are much more likely to shift in this setting, while model behavior might be correlated with the occurrence of parametric/incorrect answers in the retrieved document.

**Reasons To Accept:**

+ knowledge conflict is an important research question
+ the alternate scenario is interesting

**Reasons To Reject:**

- I'm not sure settings in previous works where the parametric answer is right and retrieved docs are wrong are "unrealistic". Retrieval could bring noisy, outdated, or irrelevant documents into LLM prompts, so it is definitely a valid setting. What could support the setting in this work, instead of claiming it to be more "realistic", is an empirical evaluation of retrieval-augmented LMs and see how often both scenarios happen. If the setting in this work is much more prevelant, it would strengthen the motivation of this work.

- According to Table 3, it seems that in the setting of focus in this work (parametric knowledge is wrong, retrieved docs are right), LLMs overwhelmingly rely on the retrieved document >95-98% of the time. This probably indicates that LLMs consistently favor the retrieved documents under these settings and we just need to improve retrieval quality, and there probably is very little to investigate about knowledge conflicts in these scenarios.

- Much of the analysis (most of Sec 4.3 and 4.4) focus on the $\mathcal{R}$ cases, i.e. the parametric answer is retained. However, it only constitute a very small fraction of cases (<5% usually). I wonder if more effort should be spent on investigating the $\mathcal{U}_i$ cases where the answer changed but still wrong.

- An important point is that we often don't know right and wrong at inference time for either parametric answers or retrieved documents. The findings and proposed "interventions" rely heavily on the assumption that "parametric is wrong but retrieved is right", which might not be applicable when the correctness is not known beforehand.

- The intervention experiments might be a bit trivial technically, simply adding or removing the parametric answer in the retrieved document. This might not be a great issue though.

- It would be nice to present some qualitative analysis and/or examples in the main paper.

---

> ### Author Rebuttal · Authors · 2024-05-30
>
> We thank the reviewer for the detailed feedback. In the following, we address the concerns:
> >I'm not sure settings in previous works [...] are "unrealistic". Retrieval could bring noisy, outdated, or irrelevant documents into LLM prompts
>
> Noisy documents are indeed realistic. However, prior work studied **synthetic** documents designed to contradict model knowledge, which is unrealistic. We study real documents, which is more realistic.
> >If the setting in this work is much more prevalent, it would strengthen the motivation of this work.
>
> RAG is often applied to **trusted** documents in OOD domains for the LM ​​[a,b,c]. Moreover, as retrieval improves, real-world systems will move closer to our setting.
> [a] https://link.springer.com/chapter/10.1007/978-3-031-60615-1_9
> [b] https://arxiv.org/abs/2309.02233
> [c] https://arxiv.org/abs/2403.16295
> >there probably is very little to investigate about knowledge conflicts in these scenarios.
>
> The parametric bias finding shows that LLMs may read conflicting contexts incorrectly. This risk is even larger for more realistic systems (App. G).
>
> >Much of the analysis [...] focus on the 𝑅  cases [...] However, it only constitute a very small fraction of cases (<5% usually). I wonder if more effort should be spent on investigating the 𝑈𝑖 cases…
>
> Apart from frequency, severity of a problem also matters.
> Answer retention (𝑅) is a severe failure of RAG. RAG is used to update model knowledge. In 𝑅, context documents did not influence the model’s answer. By analyzing these errors, we discovered the parametric bias and understood answer retention better. We'll also add an analysis of the 𝑈𝑖 cases to the revision.
> >we often don't know right and wrong at inference time for either parametric answers or retrieved documents.
>
> We propose evaluating RAG beforehand on known data to check for parametric bias in Sec. 6.
> >The findings and proposed "interventions" rely heavily on the assumption that "parametric is wrong but retrieved is right",
>
> We assume this to create a clean experimental setting. “Interventions” are experiments to study the phenomenon, not practical solutions.
> > It would be nice to present some qualitative analysis and/or examples in the main paper.
>
> Please consider:
> - Table 1: example of setup differences
> - Section 4.3.1: example of parametric answer in context
> - Section 4.4.2: example of adding parametric answer to the input.
>
> We ask the reviewer to elaborate on what other examples would strengthen our work.

---

> > ### Comment · Reviewer_2iuH · 2024-06-04
> >
> > I would like to thank the authors for the detailed response. Overall I understand why this work matters in the constrained setting of "parametric is wrong but retrieved is right", but it is often very hard to make these assumptions for retrieval-augmented LMs at inference time due to the potential noise/irrelevance of retrieval. In addition, the analysis often focuses on the most unlikely scenarios (<5%, points 2 and 3 in the review): the authors acknowledge that analysis could be expanded to more prevalent scenarios in knowledge conflicts instead of the fringe cases ("We'll also add an analysis of the 𝑈𝑖 cases to the revision"), but I'm not sure if this is entirely within the scope of camera ready revision.

---

> > > ### Author Response · Authors · 2024-06-05
> > >
> > > We thank the reviewer for the response. We would like to expand a bit on some of the points mentioned by the reviewer.
> > > >it is often very hard to make these assumptions for retrieval-augmented LMs at inference time due to the potential noise/irrelevance of retrieval
> > >
> > > The assumption of "parametric is wrong but retrieved is right" does indeed define the scope of applicability of our findings. However, ensuring correct model behavior in this setting is a **necessary** step if the goal is to ensure factual correctness using RAG. Ensuring relevance and quality of the retrieved documents is another, complementary step. **Both** of these steps are necessary for the final goal of factually correct outputs.
> > >
> > > >In addition, the analysis often focuses on the most unlikely scenarios (<5%, points 2 and 3 in the review)
> > >
> > > We think this is an important disagreement point and would like to explain our thinking and observations. We hope this provides more insight into why we focus on the R subset.
> > >
> > > Frequency is not the sole factor in determining the importance of a problem. For example, adversarial attacks rarely happen in practice, yet there is substantial research to ensure adversarial robustness of models. Analogously, the R subset may appear to be a "fringe case", but it is a well-defined problem for which we provide a deeper analysis.
> > > Out of all incorrect open-book answers, the R subset contains only the ones which did not change at all after introducing the context. This specificity, together with our analysis is what makes it actionable.
> > >
> > > Ui subset, however, includes all other incorrect answer categories "lumped" together.
> > > If you look into the Ui subset, it is not one "prevalent scenario", but rather many of them:
> > > We manually classify 100 samples in 𝑈𝑖 for the Mixtral on NQ:
> > > - 42 incorrect answer format
> > > - 10 model answer is also correct, but != GT
> > > - 13 model answer is correct while the GT is incorrect
> > > - 12 model being bad at reading tables
> > > - 13 incorrect reading (prediction appears in context, but != GT)
> > > - 4 answers not from context
> > > - 5 model wrongly claims the answer is not in the context
> > > - 3 context is illegible
> > >
> > > Analysing the Ui subset, as proposed by the reviewer, would actually require separating each sub-category. This would turn each of them into actionable, but small subsets, similar to R.
> > >
> > > >the authors acknowledge that analysis could be expanded to more prevalent scenarios
> > >
> > > We would like to clarify that we intended to add to the revision the manual analysis above for the Ui subset, together with the provided motivation for focusing on the seemingly smaller but actionable R subset.

---

### Official Review · Reviewer_bvQZ · 2024-05-13

**Rating:** 7
**Confidence:** 4
**Ethics Flag:** 1

**Summary:**

Past work on LLM knowledge conflicts has shown that models struggle to correctly use in-context passages to answer the question and fall back on the parametric answer that they have memorized. Follow-up work showed that this is less true if the context is modified sufficiently to serve as convincing evidence. However, all these approaches relied on artificial edits to the context passage.

This paper proposes that LLM behavior in knowledge conflicts should be tested when (1) the parametric answer is incorrect and (2) the correct evidence passage is provided. Authors find that even in this realistic scenario where the context is not artificially edited, LLMs again predict their parametric answer a certain percentage of the time. Additionally, a large proportion of contexts result in the LLM in predicting a different incorrect answer. This behavior is established across several QA datasets and for 3 open-source models: Llama2-7B/70B, Mistral-7B, Mixtral-8x7B.

They further study the cause of this concerning behavior. They identify a cause they name `parametric bias': the model predicts the parametric answer again when it reappears in the context even though there is a correct answer elsewhere. That is, there is a confirmation bias where models predict their parametric answer even if it occurs in unrelated contexts.

**Questions To Authors:**

1. Table 3: As stated before, what is in the error category $U_i$?
2. Table 4: Can yous separate out $P(R | a_p \subseteq c)$? Why did you choose to combine $P(R \cup U_i | a_p \subseteq c)$?
3. Sec 4.4.1: Since both $P(R)$ and $P(U_c)$ are decreasing, that means that $P(U_i)$ is increasing i.e. the model is predicting neither the parametric nor the correct contextual answer. Is the model predicting the masked (edited) parametric answer instead (thus putting it in the $U_i$ set? What do yo see is the error case?
4. Sec 4.4.1: How do you draw this conclusion? The last line is the definition of $R$ and not $U_i$. “This indicates that the examples that were influenced by the incorrect parametric answer now moved into the Incorrect update category $U_i$. *This suggests that the model might be resorting to its original answer in cases when it cannot find the correct answer in the context.*”
5. Sec 4.4.2: What explains the increase in $P(U_i)$ (in 7 of 18 settings) when we add back the parametric answer with the phrase "unrelated context"?

**Reasons To Accept:**

- The identified `parametric bias' is concerning, and they provide additional evidence in the Appendix. In particular, they find that other datasets, such as the FreshLLMs QA dataset, also contain context passages with parametric answers (in addition to the correct answer)
    - This characterization of confirmation bias is dangerous for down-stream RAG applications where users expect models to reason about the context correctly
- The `parametric bias' is established through 3 experiments
    1. The models make a mistake even with the gold context if the parametric answer appears somewhere in the passage frequently
    2. An experiment is conducted to show that masking the parametric answer from the context reduces the number of times the models retain the parametric answer (caveat discussed later)
    3. An additional experiment shows that adding the parametric answer to the context with the phrase "Unrelated context:" promotes the LLM to generate the incorrect parametric answer

**Reasons To Reject:**

- I have one major question for the authors: why did you focus on the smaller fraction of answer retention errors over the larger fraction of errors where the model predicts a different but incorrect answer?
    - This is concerning because this category of errors contributes to a larger fraction of errors overall
    - In all ablation experiments, the fraction of errors of this kind seems to be increasing but are largely ignored

---

> ### Author Rebuttal · Authors · 2024-05-30
>
> We thank the reviewer for the thorough review and answer the questions below.
>
> > why did you focus on the smaller fraction of answer retention errors [...]?
>
> Frequency isn’t the only metric for problem importance, severity also matters. E.g, adversarial attacks are rare in real world. Yet, we study them. Answer retention is a severe failure of RAG. RAG is used to update model knowledge [a]. In these errors, context documents did not influence the model’s answer. By analyzing them, we discovered the parametric bias and understood answer retention better. We'll add this to the revision.
> [a] https://arxiv.org/abs/2005.11401
> > 1. Table 3:[...] what is in the error category 𝑈𝑖?
>
> We classify 100 samples in 𝑈𝑖 for the Mixtral on NQ:
> - 42 incorrect answer format
> - 10 model answer is also correct, but != GT
> - 13 model answer is correct while the GT is incorrect
> - 12 model being bad at reading tables
> - 13 incorrect reading (prediction appears in context, but != GT)
> - 4 answers not from context
> - 5 model wrongly claims the answer is not in the context
> - 3 context is illegible
>
> We’ll add this to the appendix.
> > 2. Table 4: Can you separate out 𝑃(𝑅|𝑎𝑝⊆𝑐)? Why did you choose to combine 𝑃(𝑅∪𝑈𝑖|𝑎𝑝⊆𝑐)?
>
> 𝑅∪𝑈𝑖  includes all wrong open-book answers. We emphasize that 𝑎𝑝⊆𝑐 biases models towards wrong answers. We agree reporting only  𝑃(𝑅|𝑎𝑝⊆𝑐) would be clearer since we only analyze this type of error. We’ll change the presentation in the revision.
>
> For Llama7B, separating out  𝑃(𝑅|𝑎𝑝⊆𝑐) shows a stat. significant difference in answer retention likelihood for all datasets (+8.8*, +4.1*, +4.8*, +8.7*, +6.6*, +10.7*), emphasizing the parametric bias phenomenon
> >3. Sec 4.4.1: [...] Is the model predicting the masked (edited) parametric answer instead [...]? What do you see is the error case?
>
> ≈50% are predictions of edited answers. Others: wrong format, stated inability to answer using context, non-GT correct answers, synonyms (if also in context).
> > 4.Sec 4.4.1: How do you draw this conclusion?
>
> We apologize for the confusion: In masking experiments, some cases of answer retention remain. We suspect that models resort to the original answer not only due to the parametric bias but in some cases, due to limited reading abilities -- if parametric bias were the only reason, masking the parametric answer should eradicate answer retention.
> >5. Sec 4.4.2: What explains the increase in 𝑃(𝑈𝑖)?
>
> The majority of these cases are due to wrong answer format.

---

> ### Comment · Reviewer_bvQZ · 2024-06-06
> **Comment after rebuttal**
>
> After reading the author responses to all reviews, I am somewhat convinced by their answer to my concern.
>
> | this category of errors contributes to a larger fraction of errors overall
>
> The authors address this with two observations:
> 1. The category of errors where the model updates to a different incorrect answer is also a fragmented category. Explaining any such update would also require explanations for relatively small fractions of errors
> 2. They report a semantic matching metric (instead of EM) which further highlights that the "retention" errors are higher than initially estimated.
>
> | the fraction of errors of this kind (new incorrect answer) seems to be increasing but are largely ignored
>
> I am convinced by their breakdown of this error category.
>
> I am increasing my score from 6 -> 7.

---

### Decision · Program_Chairs · 2024-07-10

**Decision:**

Accept

**Comment:**

The paper presents an empirical study/analysis of LLM's behavior under knowledge conflict -- between model's internal parametric knowledge and provided in-context documents. The paper is clearly written with valid evaluation setting. While they study slightly newer angle to the analysis on parametric vs. non-parametric knowledge -- looking at cases where model's initial prediction was incorrect and provided in-context passage is correct. The reviewers were concerned this only account for a small subset of examples, but these are still valuable subset. The takeaways/actionable items from the analysis is a bit unclear, and the scope of analysis is a bit limited (and overlaps with rich prior work).



One point:
I agree with reviewer 2iuH that this setting is not necessarily "realistic" than prior settings, and recommend changing the title. Also --
"Re: prior work studies "synthetic document"" is not a valid representation of prior work. "Rich Knowledge Sources Bring Complex Knowledge Conflicts: Recalibrating Models to Reflect Conflicting Evidence" Chen at al EMNLP 2022, Section 5 studies knowledge conflict where real documents with conflicts are drawn from different time stamped corpus or different interpretation of the question.

[comments from the PCs] Please revise your paper to better represent prior work as the AC notes.